# 📏 Preserving LLM Capabilities through Calibration Data Curation: From Analysis to Optimization

**Bowei He[1], Lihao Yin[2], Huiling Zhen[2], Shuqi Liu[2], Han Wu[2],**
**Xiaokun Zhang[1,†], Mingxuan Yuan[2], Chen Ma[1,†]**
[1] Department of Computer Science, City University of Hong Kong, [2] Huawei, Hong Kong
[†] dawnkun1993@gmail.com, chenma@cityu.edu.hk

## Abstract

Post-training compression has been a widely employed approach to scale down large language model (LLM) and facilitate efficient inference. In various proposed compression methods, including pruning and quantization, calibration data plays a vital role by informing the weight importance and activation dynamic ranges. However, how calibration data impacts the LLM capability after compression is less explored. Few of the existing works, though recognizing the significance of this study, only investigate the language modeling or commonsense reasoning performance degradation from limited angles, like the data sources or sample amounts. More systematic research is still needed to examine the impacts on different LLM capabilities in terms of compositional properties and domain correspondence of calibration data. In this work, we aim at bridging this gap and further analyze underlying influencing mechanisms from the activation pattern perspective. Especially, we explore the calibration data's impacts on high-level complex reasoning capabilities, like math problem solving and code generation. Delving into the underlying mechanism, we find that the representativeness and diversity in activation space more fundamentally determine the quality of calibration data. Finally, we propose a calibration data curation framework based on such observations and analysis, enhancing the performance of existing post-training compression methods on preserving critical LLM capabilities. Our code is provided in Link.

## 1 Introduction

Recent large language models (LLMs) like GPT4 [Achiam et al., 2023], LLaMA2/3 [Touvron et al., 2023, Dubey et al., 2024], Qwen2.5 [Yang et al., 2024], Phi-3 [Abdin et al., 2024], have achieved the remarkable performance on human-level challenging tasks and found great application potentials. However, due to the fast expansion of model scales and energy consumptions, most of them can only be deployed to the server clusters currently. Post-training LLM compression methods like pruning [Ma et al., 2023] and quantization [Xiao et al., 2023], aiming at reducing the model memory usage and improving inference speed, have paved the way for on-device deployment of such powerful LLMs. Due to the training-free property, these methods exhibit obvious advantages on efficiency and convenience compared with other schemes like sparsity-aware training [Jaiswal et al., 2022] and quantization-aware training [Liu et al., 2024], thus gaining widespread adoption[Tan et al., 2024].

In the post-training LLM compression, calibration data serves as a small set of sampled inputs used to analyze layer activations and guide the compression process to minimize capability degradation [Williams and Aletras, 2024]. In detail, for pruning, calibration data enables the evaluation of weight importance with activation-aware metrics, helping determine which connections to remove. For quantization, this data is vital for identifying the dynamic ranges of activations, thus informing the adjustment of quantization parameters and quantization strategies regarding symmetry and granularity.

Though playing such a significant role, calibration data is less investigated by researchers whose attentions are mainly focused on developing more intricate compression strategies [Liu et al., 2025].

In fact, conventional LLM compression ways generally assume robustness to calibration data distributions and characteristics by default [Sun et al., 2024a], as shown in Figure 1. However, some recent works have noticed the calibration data variations can also bring unnegelectable influence to LLM compression performance [Wu et al., 2023, Zhang et al., 2024, Williams and Aletras, 2024]. Early works [Zhang et al., 2024, Lin et al., 2024, Jaiswal et al., 2024] explore the influence of calibration data

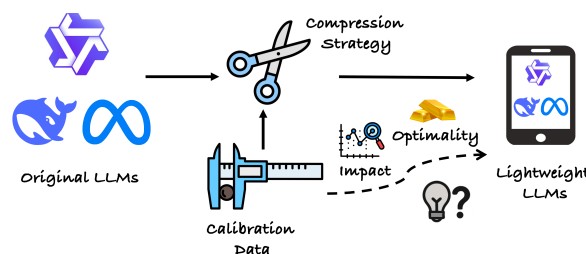

Figure 1: Calibration data in LLM compression.

amounts and concluded that additional calibration examples offer diminishing performance gains. Some further works [Williams and Aletras, 2024, Ji et al.] investigate the impact of calibration sources and found this can introduce even more significant performance variations than different compression methods. Besides, aligning the sample length between quantization calibration data and benchmark data is also regarded as important for maintaining evaluation performance [Lee et al., 2023]. Though bringing some insights, these works are still one-sided and only focus on a certain perspective of calibration data. In addition, most of them are limited to the basic language modeling perplexity and commonsense reasoning capability evaluation. Most importantly, they overlook the exploration of the underlying influence mechanism and what constitutes optimal calibration data.

To address these limitations, we first conduct a comprehensive and systematic study on the impact of calibration data for different LLM capabilities, especially high-level reasoning capabilities. In detail, we try to answer following two questions with empirical evidence: **Q1:** *How do compositional properties of calibration data influence capability preservation?* **Q2:** *How does domain correspondence of calibration data influence capability preservation?* Different from previous works that only recognize the superficial phenomenon of the sensitivity to calibration data variations [Williams and Aletras, 2024], we attempt to analyze the underlying influence mechanism when answering such two questions. Furthermore, based on these observations and analysis, we ask: **Q3:** *What is optimal calibration data for preserving critical capabilities in LLM compression?* **Q4:** *How to curate the calibration data for best preserving capabilities given available sources?* To answer them, we try to bridge the capability preservation with calibration data characteristics and then define the optimality from several dimensions. Finally, we design a curation framework by selecting and processing optimal calibration data from available sources to achieve the best capability preservation effect.

In a summary, our contributions are as three folds: 1) we conduct extensive empirical exploration on the impact of calibration data variations from different compositional properties and domain correspondence perspectives; 2) we analyze the underlying influence mechanism and point out that the activation space pattern determines the calibration data's optimality more fundamentally; 3) we propose a three-stage calibration data curation framework for optimizing the LLM capability preservation, which can be integrated with existing compression strategies due to the orthogonality. The evaluation in specific and general deployment scenarios both demonstrate its empirical advantages.

## 2 Related Works

**Large Language Model Compression** employs post-training pruning and quantization to enhance efficiency while maintaining performance. Pruning techniques span unstructured approaches [Frantar and Alistarh, 2023, Yin et al., 2024] removing individual weights; semi-structured methods [Sun et al., 2024a, Dong et al., 2024] eliminating predetermined weight blocks for hardware compatibility; and structured strategies [Ma et al., 2023, Ashkboos et al., 2024] removing entire architectural components like attention heads. Quantization reduces parameter and activation precision through weight quantization [Frantar et al., 2023, Dettmers et al., 2024, Lin et al., 2024], converting high-bit floating-point to low-bit integers; activation quantization [Xiao et al., 2023], calibrating layer output dynamic ranges; and KV cache quantization [Hooper et al., 2024], optimizing memory for extended sequence tasks. Critically, calibration data guides compression by informing pruning decisions and establishing quantization parameters. Its compositional characteristics and domain correspondence profoundly impact model capability preservation, which is the focus of our this work.

**Calibration Data** guides post-training compression by aligning with typical input distributions to preserve model capabilities. Beyond computer vision studies [Zhang et al., 2025, Shang et al., 2024], prior LLM compression research has examined calibration data from isolated perspectives: sequence lengths [Lee et al., 2023], data sources [Wu et al., 2023, Williams and Aletras, 2024, Ji et al., Khanal and Capone, 2024, Bandari et al., 2024], sample amounts [Williams and Aletras, 2024, Jaiswal et al., 2024, Zhang et al., 2024], and languages [Zeng et al., 2024, Kurz et al., 2024] for multilingual models. However, these works typically offer superficial descriptions of calibration data effects. In this work, we systematically investigate how calibration data characteristics affect capability preservation, especially the complex reasoning capabilities. Furthermore, we explore underlying influence mechanisms and develop appropriate curation strategies.

## 3 Exploring Calibration Data Variation's Impact on LLM Capabilities

### 3.1 Experiment Preparation

**Large Language Models** We employ the following two powerful and representative open-sourced LLMs to conduct the exploration: LLaMA3-8B-Instruct [Dubey et al., 2024] and Qwen2.5-7B-Instruct [Yang et al., 2024]. They are both multilingual LLMs, possessing general knowledge and a certain level of reasoning ability.

**LLM Compression Schemes** To ensure comprehensiveness, we select two representative post-training pruning methods: SparseGPT [Frantar and Alistarh, 2023] for unstructured pruning and Wanda [Sun et al., 2024a] for semi-structured pruning. The pruning ratio is set as 50% for SparseGPT, and the block pattern is set as $4:8$ for Wanda. We did not conduct experiments with structured pruning methods, considering their performance degradation is super significant under high pruning ratios. As for the post-training quantization, we choose the widely adopted GPTQ [Frantar et al., 2023] and more recent AWQ [Lin et al., 2024] with the bit number set as 4.

**Calibration Data Sources** The calibration data can be generally classified into two categories: **pre-training data** and **downstream data**. As for the first category, similar to previous works, we take the following datasets as the calibration data sources: **C4** [Raffel et al., 2020], **WikiText**, and **SlimPajama**. Except for the experiments investigating the impact of sample amounts and sequence lengths, we randomly sample 128 sequences with the token length as 2048 from corresponding sources as calibration data. As for the downstream data, like commonsense/math/code/multilingual domains, we only employ them for calibration when investigating the impact of data format and domain correspondence.

**Evaluation Benchmarks** To comprehensively evaluate the capability preservation of compressed LLMs, we take the benchmarks focusing on different capabilities, especially some high-level complex reasoning ones: 1) **Language modeling**: WikiText2 [Merity et al., 2022], PTB [Marcus et al., 1993]; 2) **Commonsense reasoning**: BoolQ [Clark et al., 2019], PIQA [Bisk et al., 2020], HellaSwag [Zellers et al., 2019], WinoGrande [Sakaguchi et al., 2021], ARC-Easy [Clark et al., 2018], ARC-Challenge [Clark et al., 2018], and OpenbookQA [Mihaylov et al., 2018]; 3) **Mathematical problem solving**: GSM8K [Cobbe et al., 2021], MATH(including subtasks like algebra, counting-and-prob, geometry, intermediate-algebra, num-theory, prealgebra, math-precalc) [Hendrycks et al.], Minerva-Math [Lewkowycz et al., 2022] with custom prompts; 4) **Code generation**: HumanEval [Chen et al., 2021], MBPP [Austin et al., 2021]; 5) **Multilingual comprehension:** ARC-Multilingual [Lai et al., 2023], HellaSwag-Multilingual [Lai et al., 2023].

### 3.2 Variation on Calibration Data Compositional Properties (Q1)

The compositional property variations of calibration data can bring non-negligible influence to the LLM capability preservation. To reveal the connection between them, we conduct the exploration from the perspectives including sequence lengths, sample amounts, data sources, and data format.

**Variation on Calibration Data Sequence Lengths** To investigate the effect of calibration data sequence lengths on capability preservation, we vary the lengths across a range of values (128, 256, 512, 1024, and 2048 tokens) while evaluating four compression methods across both LLaMA3-8B and Qwen2.5-7B models. Our experimental results, as shown in Figure 2, reveal that mathematical problem-solving shows significant sequence length sensitivity in pruning methods (SparseGPT's performance drops by 25.5% at short sequences), while code generation exhibits non-monotonic

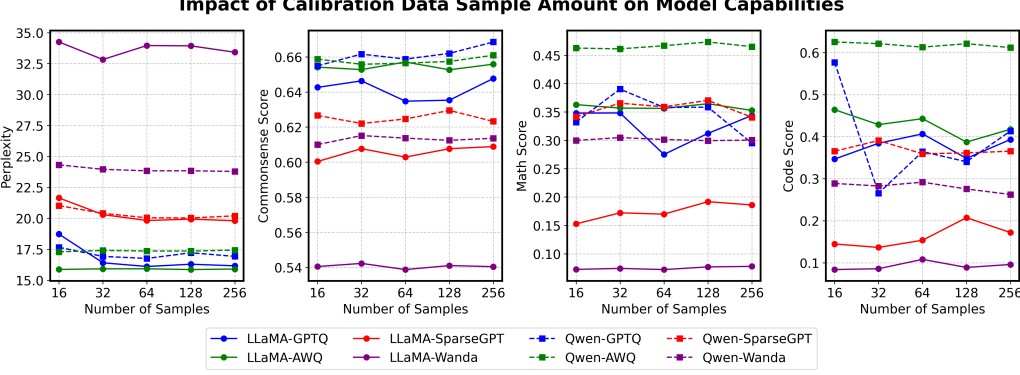

Figure 2: Impact of calibration data sequence length on capability preservation across different compression methods for LLaMA3-8B and Qwen2.5-7B. Note the varying sensitivity to sequence length across capabilities and compression methods.

Figure 3: Effect of calibration sample amount on capability preservation for LLaMA3-8B and Qwen2.5-7B models. Note the diminishing returns beyond 64-128 samples for most capabilities, with AWQ showing particular robustness to small sample sizes.

patterns across all methods (AWQ's performance varies from 38.71% to 47.53%). These effects stem from gradient estimation quality, where longer sequences provide more diverse activation patterns crucial for quantization methods like GPTQ, and task-specific representation requirements, where different capabilities have distinct optimal context windows. Pruning methods generally show greater sensitivity to sequence length than quantization approaches, with AWQ's per-channel scaling providing remarkable robustness even at short sequences.

**Variation on Calibration Data Sample Amounts** We explore the effect of varying the number of calibration samples by testing configurations with 16, 32, 64, 128, and 256 samples. Our results in Figure 3 demonstrate significant capability-dependent effects, with code generation showing unexpected patterns: for LLaMA3-8B with AWQ, code performance decreases from 46.40% at 16 samples to 38.71% at 128 samples, while Qwen2.5-7B with GPTQ shows a dramatic drop from 57.67% at 16 samples to 34.03% at 128 samples. Pruning methods show greater sensitivity to sample count, with SparseGPT on LLaMA3-8B experiencing 17.9% drop in math performance with fewer samples, while AWQ maintains consistent performance even with just 16 samples across all capabilities. These effects stem from representation diversity saturation, where additional samples provide diminishing returns after capturing core activation patterns, and calibration data quality variance, where increasing samples may introduce suboptimal examples that skew results for specialized tasks. Different compression methods exhibit varying information utilization efficiency, with AWQ extracting essential patterns from minimal samples while pruning methods require more comprehensive sampling.

**Variation on Calibration Data Sources** We examine the impact of calibration data source selection by comparing three widely used pre-training datasets: C4, Wikipedia, and SlimPajama. Figure 4 reveals that source selection significantly impacts capability preservation, often exceeding the performance difference between compression methods themselves. The most striking variations appear

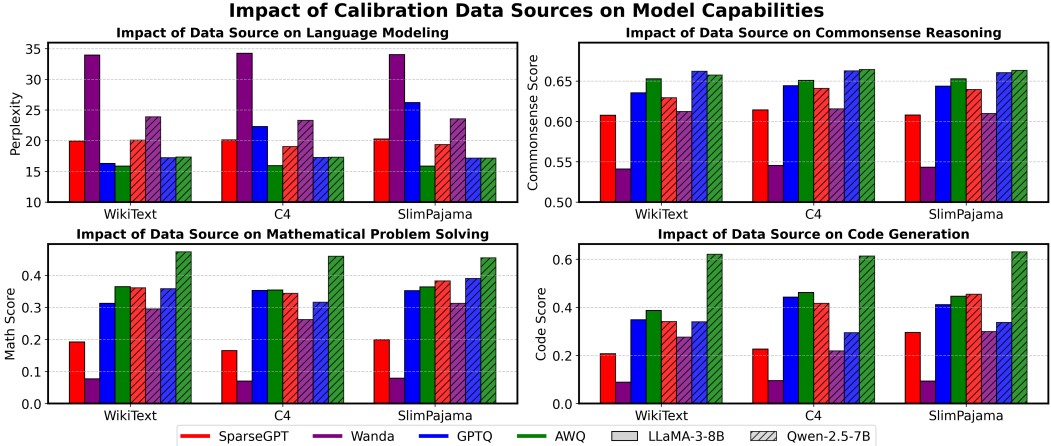

Figure 4: Impact of calibration data sources on capability preservation for LLaMA3-8B and Qwen2.5-7B. Note the significant advantage of C4 for code generation tasks, while Wikipedia provides more balanced performance across capabilities.

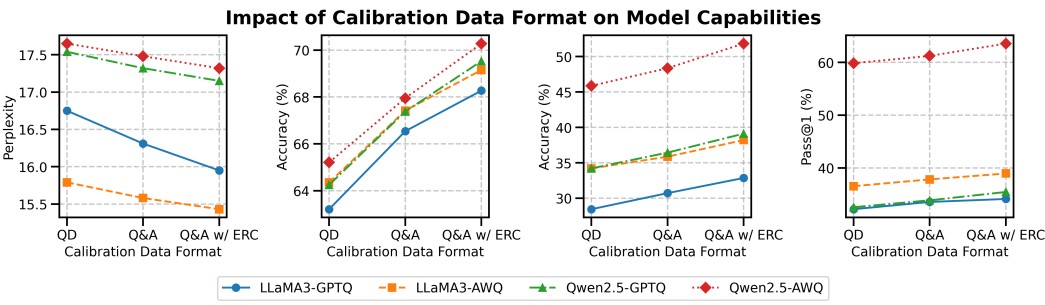

Figure 5: Impact of calibration data format on compressed LLM capability preservation.

in mathematical reasoning and code generation capabilities—for Qwen2.5-7B, SlimPajama provides an 8.7% relative improvement in math tasks over Wikipedia (38.98% vs. 35.85%), while for LLaMA3-8B, C4 dramatically improves code generation by 19.4% relative to Wikipedia (44.29% vs. 34.83%). These effects stem from domain-specific activation pattern coverage, where C4's web-crawled nature better preserves code generation capabilities while SlimPajama's diverse content maintains mathematical reasoning. Compression-specific information leverage also plays a role, with quantization and pruning methods showing different source preferences—AWQ performs best on math tasks with SlimPajama while SparseGPT achieves better results with Wikipedia. These findings highlight that strategic calibration data source selection aligned with deployment scenarios can yield substantial improvements beyond what compression method optimization alone can achieve.

**Variation on Calibration Data Format** Different from format-free continuous sequences from pre-training corpus, calibration data from downstream tasks usually follows specific formats [Bandari et al., 2024]. We investigate how different QA formats affect capability preservation using CommonseQA [1] [Talmor et al., 2019] dataset in three formats: question-only (QD), question-answer pairs (Q&A), and question-answer pairs with explicit reasoning chains (Q&A w/ ERC). Figure 5 shows that Q&A w/ ERC yields the best performance across all methods, with the most dramatic impact on complex reasoning tasks—for Qwen2.5-7B with AWQ, Q&A w/ ERC calibration achieves 51.84% on math tasks compared to 47.34% with standard pre-training data (9.5% improvement), while for LLaMA3-8B with GPTQ, it improves commonsense reasoning by 8.0% relative to question-only format (68.27% vs. 63.21%). These effects stem from reasoning pathway preservation, where explicit reasoning chains activate and maintain internal reasoning mechanisms during compression, and task-specific knowledge preservation, where detailed formats help preserve domain-specific weights

---

[1] `https://huggingface.co/datasets/tau/commonsense_qa`

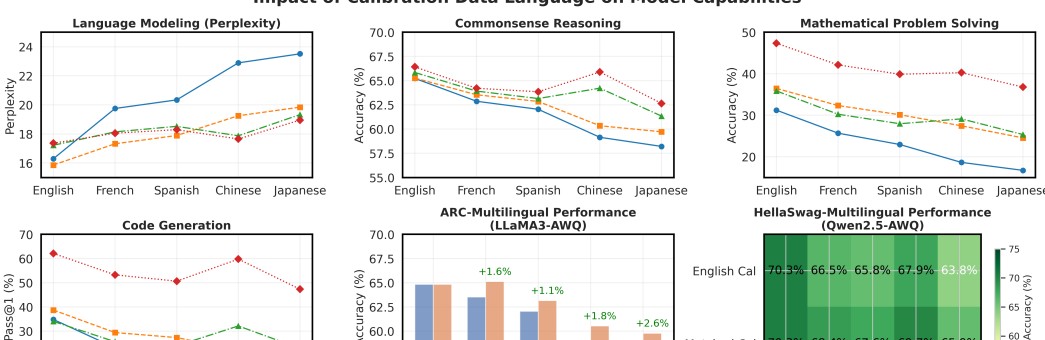

Figure 6: Impact of calibration data language on compressed LLM capability preservation.

that might otherwise be deemed less important. The substantial performance gains achievable through format optimization underscore that calibration data should include explicit reasoning chains for applications focused on reasoning tasks.

### 3.3 Variation on Calibration Data Domain Correspondence (Q2)

When the compressed LLM is intended for deployment in specific applications, downstream data is often utilized for calibration rather than pretraining data. To uncover the effect of downstream calibration data's domain variation on model capability presentation, we select three most representative perspectives to conduct this study: data languages, subjects, and reasoning difficulties.

**Variation on Calibration Data Languages** To investigate the impact of calibration data language on capability preservation, we evaluate how using data from different language subsets of C4 affects compressed model performance. Figure 6 illustrates that the most pronounced language effects appear in specialized capabilities—for mathematical tasks on LLaMA3-8B with GPTQ, English calibration data (31.22%) outperforms other languages by up to 46.4% (vs. Japanese at 16.72%), while for code generation, the performance gap reaches 62.0% (34.83% vs. 13.25%). These effects stem from alignment with pre-training distribution, as both models were primarily trained on English-dominant corpora, and language-specific activation pattern coverage, where different languages activate different regions of the model's parameter space. We also observe model-specific differences, with Qwen2.5-7B showing better resilience to non-English calibration (only 3.8% gap between English and Chinese for code generation vs. 54.3% for LLaMA3-8B). Notably, on multilingual benchmarks, using calibration data in the same language as the evaluation yields better performance than English calibration for that specific language, suggesting that for language-specific applications, matching calibration language to target language can be more effective than defaulting to English.

**Variation on Calibration Data Subjects** To investigate how calibration data from different subjects affects capability preservation, we calibrate models using three subject-specific datasets: CommonsenseQA [Talmor et al., 2019] for commonsense reasoning, MathQA[2] for mathematical problem solving, and CodeQA [3] for code generation. Table 1 shows that subject-specific calibration significantly enhances matching capabilities—MathQA calibration boosts mathematical performance by 5.92 percentage points for quantization methods and 4.35 points for pruning methods, while CodeQA enhances code generation by 7.49 points for quantization and 4.28 points for pruning. However, this enhancement comes at the cost of increased perplexity, with MathQA causing the largest degradation (average increase of 2.73 points for quantization and 3.52 points for pruning). The relative improvement is more pronounced for pruning methods—CodeQA improves code generation by up to 29.7% for SparseGPT compared to 17.9% for AWQ. These effects stem from activation pathway preservation, where calibration data activates and helps preserve subject-specific neural pathways

---

[2]https://huggingface.co/datasets/allenai/math_qa
[3]lissadesu/code_qa_updated

Table 1: Impact of calibration data subject on capability preservation. Lower perplexity and higher accuracy for other capabilities are better. Best performing subject for each capability is in **bold**.

| Compression | Calibration Data Subject | LLaMA3-8B | | | | Qwen2.5-7B | | | |
|---|---|---|---|---|---|---|---|---|---|
| | | PPL↓ | CS↑ | Math↑ | Code↑ | PPL↓ | CS↑ | Math↑ | Code↑ |
| SparseGPT (50%) | WikiText | **20.15** | 41.85 | 19.18 | 15.34 | **21.54** | 42.23 | 17.85 | 13.45 |
| | CommonsenseQA | 22.42 | **45.23** | 17.45 | 13.76 | 23.85 | **46.82** | 16.24 | 12.31 |
| | MathQA | 23.87 | 39.62 | **23.56** | 14.21 | 24.72 | 40.58 | **22.26** | 12.87 |
| | CodeQA | 23.14 | 38.95 | 16.82 | **19.28** | 24.15 | 40.13 | 15.73 | **17.46** |
| Wanda (4:8) | WikiText | **33.41** | 40.87 | 18.44 | 14.72 | **32.84** | 40.21 | 17.32 | 12.97 |
| | CommonsenseQA | 35.26 | **44.36** | 16.83 | 13.25 | 34.96 | **44.73** | 15.85 | 11.84 |
| | MathQA | 36.82 | 38.75 | **22.94** | 13.86 | 36.21 | 38.42 | **21.74** | 12.15 |
| | CodeQA | 36.15 | 37.94 | 16.21 | **18.52** | 35.92 | 38.06 | 15.26 | **16.82** |
| GPTQ (4-bit) | WikiText | **16.29** | 65.23 | 31.22 | 34.83 | **17.22** | 65.84 | 35.85 | 34.03 |
| | CommonsenseQA | 18.54 | **68.75** | 28.36 | 31.42 | 19.36 | **71.25** | 33.75 | 31.56 |
| | MathQA | 19.87 | 63.21 | **36.92** | 32.18 | 20.14 | 64.36 | **42.36** | 32.41 |
| | CodeQA | 19.34 | 62.84 | 29.75 | **39.26** | 19.58 | 63.78 | 32.61 | **42.85** |
| AWQ (4-bit) | WikiText | **15.86** | 65.26 | 36.46 | 38.71 | **17.36** | 66.42 | 47.34 | 62.10 |
| | CommonsenseQA | 17.23 | **69.37** | 34.21 | 36.22 | 18.53 | **72.86** | 45.21 | 58.64 |
| | MathQA | 18.41 | 64.15 | **41.85** | 35.89 | 19.25 | 65.38 | **54.42** | 57.21 |
| | CodeQA | 18.26 | 63.92 | 33.42 | **44.62** | 18.95 | 64.97 | 43.85 | **68.73** |

during compression, particularly important for pruning methods that directly eliminate weights rather than approximating them (see Appendix for visualization). These findings suggest pruning methods require more precise subject alignment in calibration data, while quantization methods may benefit from mixed-subject calibration for balanced capability preservation.

In addition, we also explored the effect of calibration data's reasoning difficulty to the critical LLM capability preservation, detailed in Appendix D.

# 4 Optimizing Capability Preservation with Calibration Data Curation

## 4.1 Discussion on Calibration Data Optimality for Capability Preservation (Q3)

Based on our above empirical explorations, we characterize optimal calibration data through two key dimensions that affect capability preservation. **Compositional Properties** show that ① longer sequences generally improve performance, ② sample amounts exhibit diminishing returns beyond 64-128 samples, ③ data sources impact capabilities differently (C4 excels for code, SlimPajama for math), and ④ explicit reasoning format benefits preserving reasoning capability. **Domain Correspondence** reveals that ⑤ matching calibration data language to the deployment language is crucial for multilingual applications, ⑥ subject-aligned data significantly enhances target capabilities (e.g., MathQA improves math performance by 5.92 points for quantization), and ⑦ mixed difficulty calibration provides optimal balance between specialized reasoning and general performance.

These dimensions ultimately reflect a deeper mechanism: **Representativeness and Diversity in Activation Space**. Representativeness concerns how well calibration samples trigger activation patterns typical of the target domain—explaining why domain-matched data preserves corresponding capabilities so effectively. Diversity involves the breadth of unique activation patterns triggered—demonstrated by the success of higher sample amount, mixed difficulty, and format richness with explicit reasoning chains. We thus define optimal calibration data as *a strategically curated set of samples that maximizes both representativeness and diversity in the model's activation space, with the balance determined by deployment requirements*. Actually, we also provide the internal influence mechanism analyze from spectral perspective in Appendix E to further demonstrate this point.

## 4.2 Calibration Data Curation Framework (Q4)

Based on our analysis of how calibration data characteristics affect capability preservation and above discussion on calibration data optimality, we propose a three-stage framework for **C**urating **O**ptimal **LL**M compression c**A**libration data, named as **COLA**.

**Stage 1: Dataset Selection (Domain Correspondence)**  The first stage focuses on selecting source datasets that align with the target deployment domain. This involves analyzing whether the compressed model is intended for general-purpose use or specialized tasks. For general-purpose deployment, we recommend selecting a balanced mix of pre-training datasets (e.g., WikiText for language modeling, SlimPajama for commonsense reasoning, C4 for code generation). For targeted deployment, domain-matched datasets should be prioritized (e.g., MathQA for mathematical problem solving applications). Language alignment is crucial in this stage, where datasets in the primary language of intended use should be selected. For multilingual applications, calibration data in each target language should be included, with proportions reflecting deployment priorities. Subject coverage must ensure all critical domains required for deployment are represented. Reasoning difficulty is also determined at this stage. Based on our experiments, mixed difficulty provides the best balance for general-purpose deployment, while hard samples may be preferable for specialized reasoning applications. The dataset selection can be formulated as an optimization problem:

$$S = \arg\max_{S \subseteq \mathcal{D}} \sum_{c \in C} w_c \cdot \text{coverage}(S, c), \tag{1}$$

where $S$ is the selected dataset, $\mathcal{D}$ is the pool of available datasets, $C$ is the set of target capabilities, $w_c$ is the importance weight for capability $c$, and $\text{coverage}(S, c)$ measures how well dataset $S$ covers capability $c$.

**Stage 2: Dataset Processing (Compositional Properties)**  The second stage optimizes the compositional properties of the selected datasets. This includes sequence length optimization, where datasets are processed to generate sequences of appropriate length (typically 2048 tokens for most methods, but adaptable based on the specific compression method). For AWQ, which showed robustness to sequence length variation, shorter sequences (512-1024 tokens) may be sufficient. Format enrichment is another key processing step. For datasets lacking structural richness, we enhance them by converting to Q&A format with explicit reasoning chains where possible. This involves identifying implicit reasoning in text passages, reformulating as question-answer pairs, and adding intermediate reasoning steps when beneficial.

**Stage 3: Sample Selection (Representativeness and Diversity in Activation Space)**  The final stage selects individual samples to maximize representativeness and diversity in activation space. This begins with activation pattern extraction, where for a candidate pool of processed samples, we run a forward pass through the uncompressed model and extract layer-wise results. For each sample $x_i$, we obtain an activation vector $\mathbf{a}_i = \left[\mathbf{h}_i^1, \mathbf{h}_i^2, \ldots, \mathbf{h}_i^L\right]$, where $\mathbf{h}_i^l$ represents the aggregated hidden state activations at layer $l$ for sample $x_i$. We then apply dimensionality reduction to the activations using random projection which exhibits obvious efficiency advantages [Bingham and Mannila, 2001]:

$$\mathbf{a}_i' = \frac{1}{\sqrt{d}}\mathbf{R}\mathbf{a}_i, \quad \{C_1, C_2, \ldots, C_k\} = \text{k-means}(\{\mathbf{a}_1', \mathbf{a}_2', \ldots, \mathbf{a}_n'\}, k), \tag{2}$$

where $\mathbf{R}$ is a $d \times D$ random matrix with entries drawn from a standard normal distribution, $D$ is the original dimension, $d$ is the reduced dimension ($d \ll D$), $k$ is the target number of clusters. From each cluster $C_j$, we select samples closest to the centroid to form the final calibration set:

$$x_j^* = \arg\min_{x_i \in C_j} \|\mathbf{a}_i' - \mu_j\|_2, \tag{3}$$

where $\mu_j$ is the centroid of cluster $C_j$. These samples from all clusters $C_1, C_2, ...C_k$ ensures the diversity in the activation space, while the closeness to the centroid in each cluster guarantees the representativeness. Note the number of clusters $k$ directly controls the final calibration sample amount, allowing precise adjustment based on the compression method's requirements—smaller yet diverse sets for methods like AWQ that show robustness to sample quantity, and larger sets for pruning methods that exhibit higher sensitivity to calibration data amount.

By systematically addressing dataset-level domain correspondence, compositional characteristics, and ultimately sample-level activation space representation, our curation framework produces compact, high-quality calibration datasets that maximize capability preservation for LLM compression across diverse deployment scenarios.

Table 2: Performance comparison of different calibration data approaches on general deployment scenario. The best performing approach under each capability is in **bold**.

| Compression | Calibration Data | LLaMA3-8B | | | | Qwen2.5-7B | | | |
|---|---|---|---|---|---|---|---|---|---|
| | | PPL↓ | CS↑ | Math↑ | Code↑ | PPL↓ | CS↑ | Math↑ | Code↑ |
| AWQ (4-bit) | WikiText (random) | 15.86 | 65.26 | 36.46 | 38.71 | 17.36 | 66.42 | 47.34 | 62.10 |
| | C4 (random) | 15.48 | 66.21 | 37.19 | 39.87 | 17.00 | 67.42 | 48.29 | 63.72 |
| | SlimPajama (random) | 15.55 | 66.53 | 37.36 | 39.48 | 17.08 | 67.75 | 48.46 | 63.28 |
| | Self-Gen | 15.59 | 67.08 | 37.51 | 39.75 | 17.12 | 68.04 | 48.66 | 63.67 |
| | COLA | **15.41** | **67.42** | **37.85** | **40.17** | **16.95** | **68.47** | **49.02** | **64.15** |
| SparseGPT (50%) | WikiText (random) | 20.15 | 41.85 | 19.18 | 15.34 | 21.54 | 42.23 | 17.85 | 13.45 |
| | C4 (random) | 19.36 | 42.67 | 19.64 | 15.88 | 20.85 | 43.12 | 18.24 | 13.87 |
| | SlimPajama (random) | 19.58 | 42.81 | 19.78 | 15.77 | 20.98 | 43.29 | 18.37 | 13.80 |
| | Self-Gen | 19.65 | 43.61 | 19.85 | 15.92 | 21.07 | 43.92 | 18.42 | 13.92 |
| | COLA | **19.31** | **44.23** | **20.12** | **16.14** | **20.72** | **44.47** | **18.65** | **14.10** |
| Wanda (4:8) | WikiText (random) | 33.41 | 40.87 | 18.44 | 14.72 | 32.84 | 40.21 | 17.32 | 12.97 |
| | C4 (random) | 32.62 | 41.52 | 18.82 | 15.14 | 32.15 | 40.83 | 17.66 | 13.31 |
| | SlimPajama (random) | 32.83 | 41.65 | 18.98 | 15.06 | 32.24 | 40.92 | 17.83 | 13.24 |
| | Self-Gen | 32.87 | 42.61 | 19.07 | 15.28 | 32.32 | 41.82 | 17.87 | 13.42 |
| | COLA | **32.14** | **43.15** | **19.36** | **15.46** | **31.65** | **42.34** | **18.15** | **13.59** |
| GPTQ (4-bit) | WikiText (random) | 16.29 | 65.23 | 31.22 | 34.83 | 17.22 | 65.84 | 35.85 | 34.03 |
| | C4 (random) | 15.93 | 66.15 | 31.87 | 35.80 | 16.86 | 66.89 | 36.50 | 34.92 |
| | SlimPajama (random) | 16.01 | 66.47 | 32.00 | 35.56 | 16.95 | 67.17 | 36.68 | 34.68 |
| | Self-Gen | 16.03 | 67.14 | 32.22 | 35.91 | 16.98 | 67.74 | 36.89 | 35.02 |
| | COLA | **15.83** | **67.52** | **32.56** | **36.18** | **16.79** | **68.15** | **37.23** | **35.22** |

## 4.3 Empirical Performance Evaluation

We evaluate our calibration data curation framework across two settings: general deployment and targeted deployment. For general deployment, we compare against random samples from standard pre-training datasets (WikiText, C4, SlimPajama). Besides, we consider the recent Self-Generating then Sampling (Self-Gen) baseline [Ji et al.]. As for the targeted deployment, we provide our settings and results in Appendix F due to the space limitation. More details regarding the implementation of our proposed COLA framework can be seen in Appendix G. In general deployment scenarios (Table 2), our activation-aware curation framework consistently outperforms both random sampling and Self-Gen approaches across all capabilities and compression methods. For LLaMA3-8B with SparseGPT, our approach achieves 44.23% on commonsense reasoning tasks compared to 41.85% for WikiText random sampling and 43.61% for the Self-Gen approach. While the absolute improvements appear modest (around 1-2 percentage points), they are consistent across different models, capabilities, and compression methods. The improvements are particularly noticeable for pruning methods (SparseGPT and Wanda), which aligns with our earlier observation that pruning methods exhibit higher sensitivity to calibration data quality. Besides, the observation in targeted deployment scenarios (Appendix F) also demonstrates the effectiveness our COLA.

The performance gains from our curation framework demonstrate that strategically selecting calibration data based on activation patterns provides significant benefits for capability preservation in compressed LLMs. By optimizing both representativeness and diversity in the activation space, our approach successfully preserves critical capabilities and achieves better performance on corresponding evaluations. These results validate our hypothesis that the efficacy of calibration data stems from how it activates the model's parameter space rather than solely from its observable characteristics.

## 5 Conclusions and Future Works

This work systematically investigates calibration data's impact on LLM capability preservation through compositional properties and domain correspondence. We analyze the underlying mechanisms from the activation pattern perspective, further finding that representativeness and diversity in activation space fundamentally determine calibration data optimality. Based on these observations and analysis, we develop a three-stage framework for curating optimal calibration data from available sources. Its effectiveness is validated across both general and targeted deployment scenarios. Future work will focus on developing compression method-specific calibration data curation strategies that account for algorithmic characteristics and the intrinsic properties of target LLMs.

## Acknowledgments and Disclosure of Funding

This work was supported by the Early Career Scheme (No. CityU 21219323) and the General Research Fund (No. CityU 11220324) of the University Grants Committee (UGC), the NSFC Young Scientists Fund (No. 9240127), and the Donation for Research Projects (No. 9220187 and No. 9229164).

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

# Contents

## A  Comparison of calibration data in LLM pruning and quantization

To facilitate understanding the role and effect of calibration data in LLM pruning and quantization, we highlight their difference in Table 3.

Table 3: Comparison of calibration data in LLM pruning and quantization.

| Task | Pruning | Quantization |
|---|---|---|
| Goal | Remove redundant weights, reduce model size/computation | Reduce precision of weights/activations, decrease storage/computation costs |
| Role of Calibration Data | Evaluate weight importance (e.g., via gradients, activation sparsity) | Determine dynamic ranges, adjust quantization parameters and strategies |
| Core Method | Sensitivity analysis | Statistical distribution analysis (e.g., min-max, KL divergence) |

## B  Recapping the LLM Compression Pipelines

To facilitate understanding the context of our this work, we recap the LLM compression pipeline here. In fact, the post-training LLM quantization (e.g., GPTQ [Frantar et al., 2023] and AWQ [Lin et al., 2024]) and pruning (e.g., SparseGPT [Frantar and Alistarh, 2023] and Wanda [Sun et al., 2024a]) methods can be jointly concluded as solving a layer-wise reconstruction problem:

$$\operatorname{argmin}_{\widehat{\mathbf{W}}_l} \|\mathbf{W}_l \mathbf{X}_l - \widehat{\mathbf{W}}_l \mathbf{X}_l\|_F. \tag{4}$$

Here, the Frobenius norm is taken as the objective function. The $\mathbf{W}_l$ indicate the weights corresponding to a linear layer $l$ and the $\mathbf{X}_l$ denote the layer input corresponding to the calibration data $S$ running through the model. $\widehat{\mathbf{W}}_l$ is the compressed version of $\mathbf{W}_l$. $S$ is generally a subset of LLM pre-training datasets (e.g., WikiText and C4 [Raffel et al., 2020]) for general deployment, or domain-specific datasets for targeted deployment.

In typical LLM compression pipelines, the preparation of calibration data $S$ follows a standardized process. The pipeline first loads the calibration datasets and processes them through the model's tokenizer with appropriate error handling mechanisms. A critical preprocessing step involves proper handling of special tokens, particularly ensuring the pad_token is properly defined, typically defaulting to the eos_token if undefined. For generating calibration samples, the pipeline concatenates the text data with delimiters and tokenizes it into a continuous sequence of token IDs. Fixed-length segments are then extracted through random sampling within the valid range of the input sequence, ensuring each calibration sample maintains proper context and structure.

## C  Complementary Introduction to Experiment Preparation

**Pre-training Corpus as Calibration Data Sources**   Here, we make a more detailed introduction to the three pre-training corpus as calibration data sources in our exploration: **C4** [4] [Raffel et al., 2020] is a massive web-text dataset derived from filtered Common Crawl snapshots, widely adopted for pre-training general-purpose language models due to its broad domain coverage and rigorous deduplication; **WikiText** [5] provides curated, multilingual encyclopedic text with structured semantic relationships, making it a foundational resource for knowledge-intensive NLP tasks and cross-lingual model training; **SlimPajama** [6] is a refined, deduplicated version of the RedPajama[Weber et al.] dataset, streamlining its multi-source composition (scientific papers, books, code, and web content) through rigorous filtering and standardized preprocessing to enhance efficiency in large-scale language model pre-training.

---

[4] https://huggingface.co/datasets/allenai/c4
[5] https://huggingface.co/datasets/Salesforce/wikitext
[6] https://huggingface.co/datasets/DKYoon/slimpajama-200k

Table 4: Impact of calibration data reasoning difficulty on capability preservation. Values show relative performance change (%) compared to WikiText baseline calibration. For perplexity, negative values indicate improvement; for other metrics, positive values indicate improvement. Best performing difficulty level for each capability is highlighted in **bold**.

| Compression | Reasoning Difficulty | LLaMA3-8B | | | | Qwen2.5-7B | | | |
|---|---|---|---|---|---|---|---|---|---|
| | | PPL↓ | CS↑ | Math↑ | Code↑ | PPL↓ | CS↑ | Math↑ | Code↑ |
| SparseGPT (50%) | Easy | +5.8 | +2.1 | +1.8 | +0.7 | +4.9 | +2.3 | +1.5 | +0.5 |
| | Medium | +9.4 | +3.6 | +5.2 | +2.4 | +8.3 | +4.1 | +4.8 | +2.2 |
| | Hard | +12.1 | **+5.7** | **+8.4** | **+3.5** | +11.5 | **+6.2** | **+7.9** | **+3.2** |
| | Mixed | **+4.2** | +4.9 | +7.2 | +3.1 | **+3.8** | +5.3 | +6.8 | +2.9 |
| Wanda (4:8) | Easy | +3.6 | +1.9 | +1.5 | +0.6 | +3.2 | +2.2 | +1.2 | +0.4 |
| | Medium | +6.3 | +3.2 | +4.7 | +2.1 | +5.8 | +3.8 | +4.3 | +1.9 |
| | Hard | +9.7 | **+5.3** | **+7.8** | **+3.2** | +8.9 | **+5.9** | **+7.4** | **+2.8** |
| | Mixed | **+2.8** | +4.5 | +6.7 | +2.8 | **+2.4** | +5.0 | +6.3 | +2.5 |
| GPTQ (4-bit) | Easy | +3.4 | +0.7 | +0.9 | +0.5 | +3.2 | +0.8 | +0.7 | +0.4 |
| | Medium | +5.1 | +1.8 | +4.3 | +1.8 | +4.7 | +2.0 | +3.9 | +1.6 |
| | Hard | +8.2 | +2.9 | **+7.1** | **+3.0** | +7.8 | +3.2 | **+6.8** | **+2.7** |
| | Mixed | **+2.1** | **+3.4** | +6.2 | +2.6 | **+1.9** | **+3.6** | +5.9 | +2.3 |
| AWQ (4-bit) | Easy | +2.1 | +0.5 | +0.7 | +0.3 | +1.9 | +0.6 | +0.5 | +0.2 |
| | Medium | +3.7 | +1.5 | +3.8 | +1.5 | +3.3 | +1.7 | +3.5 | +1.3 |
| | Hard | +6.8 | +2.5 | **+6.7** | **+2.8** | +6.2 | +2.8 | **+6.2** | **+2.5** |
| | Mixed | **+1.5** | **+3.1** | +5.8 | +2.4 | **+1.2** | **+3.4** | +5.4 | +2.2 |

**Implementation Details** For all experiments in this work, we use the Ubuntu 22.04 LTS system, Python 3.11.11 environment, and vLLM 0.7.2 library[7] for LLM local inference of both LLaMA3-8B-Instruct and Qwen2.5-7B-Instruct. For vLLM inference hypermeters, we set the max_tokens to 1024, temperature to 0.7, top_k to 50, top_p to 0.7, and repetition_penalty to 1. We run all experiments on a server with 128 Intel Xeon Platinum 8538 CPU @ 2.60GHz and 8 Nvidia RTX 6000 Ada GPU having 48 GB GDDR6 VRAM. We utilize official checkpoints from HuggingFace: LLaMA3-8B-Instruct[8] and Qwen2.5-7B-Instruct[9]. The evaluation part is based on the open-source repository `lm-evaluation-harness`[10], v0.4.7 version. As for the hyperparameter setting of LLM compression methods, we directly follow their original papers which are also detailed in Appendix A. Each experiment is performed for five times with different seeds and then reports the averaged performance to mitigate the randomness. Statistical significance is also assessed using two-tailed independent t-tests, with results considered significant when $p < 0.01$. For two multilingual benchmarks utilized in our experiments, they are only performed when evaluating the impact of calibration data's language. Different from other capabilities, code generation correctness is evaluated with the pass@1 rate. The three math benchmarks and code benchmark MBPP are evaluated with 4-shot and 3-shot manner, respectively, while others are evaluated with 0-shot manner. For the ease of understanding, without specified, the performance under each capability presented in the main text is the average among corresponding benchmarks. For example, the reported code scores (pass@1) are actually the average of HumanEval and MBPP. We provide the code repository for our proposed COLA framework in `https://github.com/BokwaiHo/COLA.git`.

# D  Variation on Calibration Data Reasoning Difficulties

To investigate how the difficulty level of calibration data impacts capability preservation, we stratify samples from Easy2Hard-Bench [11] [Ding et al., 2024] into three difficulty tiers including Easy(0-0.33), Medium(0.33-0.67), Hard(0.67-1.0) and create a Mixed set containing samples from all tiers. Table 4 reveals that harder samples yield greater improvements in reasoning tasks but cause larger perplexity degradation—Hard calibration improves mathematical reasoning by up to 8.4% for

---

[7] `https://github.com/vllm-project/vllm`

[8] `https://huggingface.co/meta-llama/Meta-Llama-3-8B-Instruct`

[9] `https://huggingface.co/Qwen/Qwen2.5-7B-Instruct`

[10] `https://github.com/EleutherAI/lm-evaluation-harness`

[11] `furonghuang-lab/Easy2Hard-Bench`

SparseGPT but increases perplexity by 12.1%. Meanwhile, Mixed difficulty calibration provides the most balanced preservation, achieving near-optimal improvements for reasoning tasks (within 1.5% of Hard calibration's gains) while minimizing perplexity degradation (only 1.5-4.2% increase versus 6.2-12.1% for Hard). Sensitivity to calibration difficulty varies significantly by method, with pruning methods showing up to $2.3\times$ higher sensitivity than quantization methods. These effects stem from representational richness, where hard samples activate more complex reasoning pathways that compression methods must preserve, and activation diversity, where mixed calibration benefits from complementary patterns that better maintain both reasoning and language capabilities. These findings suggest reasoning-critical applications should use Hard or Mixed calibration, while general-purpose applications should favor Mixed calibration to balance capability improvements with minimal perplexity degradation.

## E  Internal Influence Mechanism Analysis - A Spectral Perspective

To further understand the underlying mechanism behind calibration data variations' impact on capability preservation, we conducted a layer-wise spectral analysis of feed-forward networks (FFNs) weights before and after compression [Bai and Silverstein, 2010]. This analysis reveals how different calibration strategies affect the frequency domain representation of model parameters, which directly influences capability preservation. The FFNs in transformer layers are known to be the primary knowledge storage components in LLMs [Dai et al., 2022]. Our spectral analysis decomposes the FFN weights into frequency components to visualize how compression affects different frequency bands. This approach is motivated by frequency-domain interpretation of large language models, where low-frequency components (0-0.2) typically correspond to general language modeling capabilities, mid-frequency components (0.2-0.6) to common knowledge and pattern recognition, and high-frequency components (0.6-1.0) to specialized reasoning capabilities like mathematics and code generation.

We take the GPTQ (4-bit) as the example compression method for visualization. Figure 7 shows the spectral analysis of FFN weights across all 32 layers of the LLaMA3-8B model with Fourier Transformation. When comparing the original model with models compressed using different calibration data, we observe two critical patterns. First, when calibration data lacks representativeness (red line), it results in well-preserved low-frequency components but non-uniform compression and distortion in the mid-frequency bands. This explains why models compressed with random calibration data often maintain basic language modeling capabilities but show degraded performance on commonsense reasoning tasks that rely on these mid-frequency components. Second, when calibration data lacks diversity, we observe loss of high-frequency information and frequency band energy redistribution, manifested as truncation of the spectral tail and energy migration toward lower frequencies. This directly corresponds to the degradation of high-level reasoning capabilities like mathematics and code generation, which rely heavily on high-frequency components.

Figure 8 provides a more direct visualization of compression quality by showing the ratio of compressed to original magnitude across frequency bands. Our COLA approach (green line) maintains a ratio closer to 1.0 across all frequency bands, indicating more faithful preservation of the original information. In contrast, random calibration (red line) shows greater deviation from the ideal ratio, particularly in mid and high-frequency bands. This deviation is especially pronounced in deeper layers (e.g., layers 25-32), which are often responsible for higher-level reasoning capabilities in transformer models. The compression ratio analysis reveals an important insight: the most significant differences between COLA and random calibration occur in the high-frequency region (0.6-1.0), especially in the middle and deeper layers of the network. This explains why mathematical reasoning and code generation capabilities—which rely on these high-frequency components—show the most dramatic improvements when using our COLA framework compared to random calibration.

To quantify these observations, Figure 9 shows the normalized energy distribution across frequency bands for each layer. When using COLA (green line), the energy distribution more closely tracks the original model (black line) across all frequency bands, particularly in the high-frequency region. In contrast, random calibration (red line) shows excess energy in low-frequency bands and energy loss in high-frequency bands, particularly in deeper layers where complex reasoning capabilities are typically encoded. The energy distribution analysis further confirms that the high-frequency components experience the most significant energy loss during compression with random calibration data. This energy redistribution explains why models compressed with suboptimal calibration data

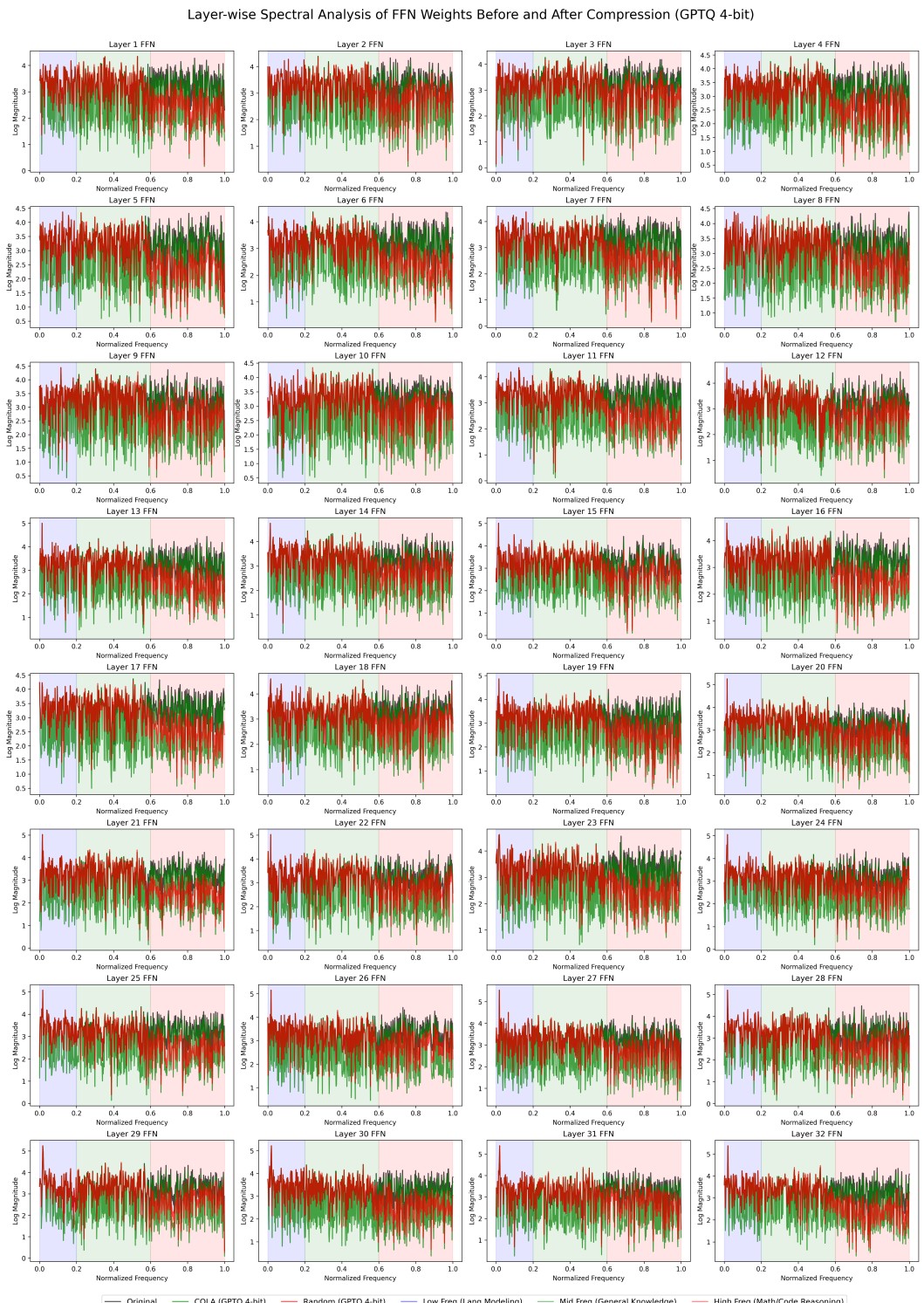

Figure 7: Layer-wise spectral analysis of FFN weights before and after compression using GPTQ (4-bit) quantization. The frequency spectrum is divided into low (0-0.2, language modeling), mid (0.2-0.6, general knowledge), and high (0.6-1.0, specialized reasoning) frequency bands.

Layer-wise Compression Ratio Analysis in Frequency Domain (GPTQ 4-bit)

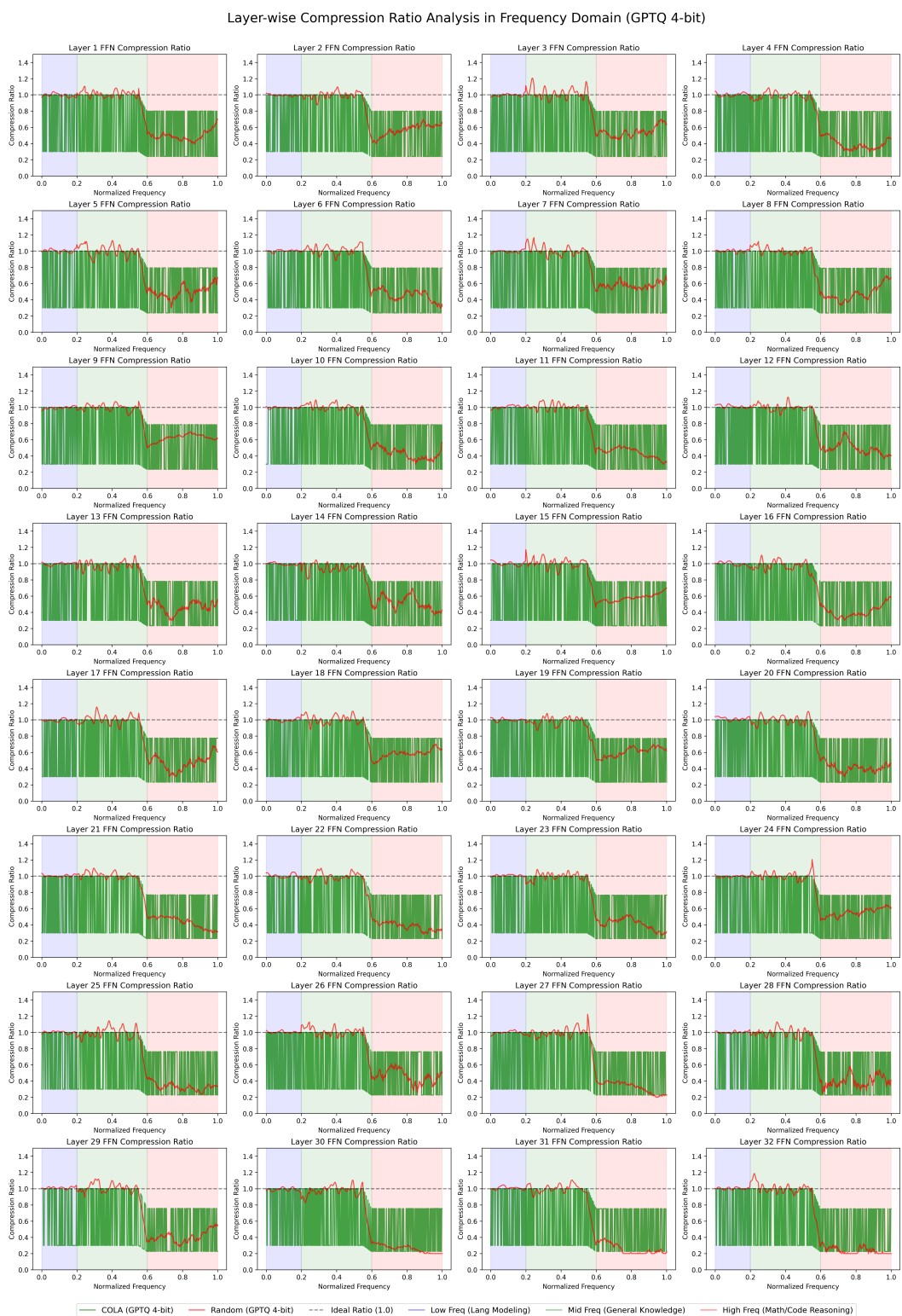

Figure 8: Layer-wise compression ratio analysis in frequency domain for GPTQ (4-bit) quantization. Values near 1.0 indicate perfect preservation of the original frequency components, while lower values indicate information loss and higher values indicate distortion.

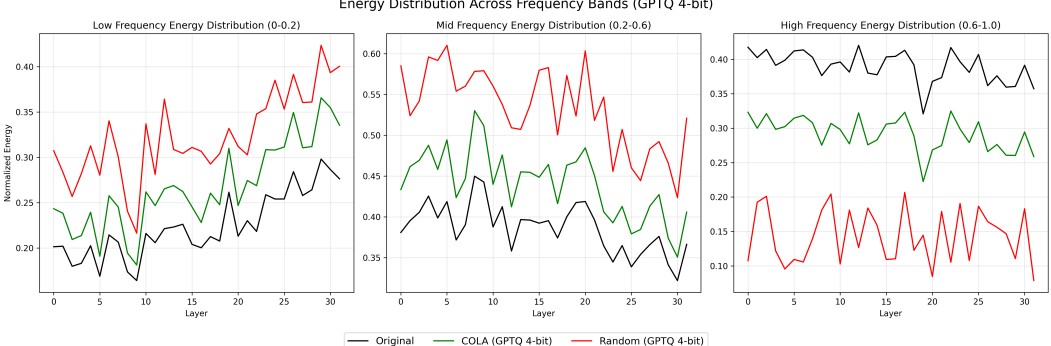

Figure 9: Energy distribution across frequency bands by layer for GPTQ (4-bit) quantization. The plots show how energy is preserved or redistributed across low, mid, and high-frequency bands after compression with different calibration approaches.

Table 5: Performance comparison of different calibration data approaches for targeted deployment scenarios. GPQ is taken as the compression scheme.

| Deployment | Calibration Data | LLaMA3-8B | | | | Qwen2.5-7B | | | |
|---|---|---|---|---|---|---|---|---|---|
| | | PPL↓ | CS↑ | Math↑ | Code↑ | PPL↓ | CS↑ | Math↑ | Code↑ |
| Math | MathQA (random) | 19.87 | 63.21 | 36.92 | 32.18 | 20.14 | 64.36 | 42.36 | 32.41 |
| | Self-Gen | 19.65 | 63.84 | 37.65 | 32.78 | 19.96 | 64.98 | 43.17 | 32.95 |
| | COLA | **19.43** | **64.15** | **38.92** | **33.12** | **19.74** | **65.42** | **44.62** | **33.27** |
| Code | CodeQA (random) | 19.34 | 62.84 | 29.75 | 39.26 | 19.58 | 63.78 | 32.61 | 42.85 |
| | Self-Gen | 19.28 | 63.25 | 30.21 | 40.05 | 19.49 | 64.13 | 33.18 | 43.76 |
| | COLA | **19.05** | **63.67** | **30.64** | **41.73** | **19.24** | **64.52** | **33.64** | **45.28** |
| Commonsense | CommonseQA (random) | 18.72 | 64.87 | 30.24 | 33.45 | 18.95 | 65.98 | 34.26 | 33.82 |
| | Self-Gen | 18.65 | 65.32 | 30.65 | 33.92 | 18.83 | 66.57 | 34.75 | 34.24 |
| | COLA | **18.41** | **65.94** | **31.12** | **34.26** | **18.62** | **67.32** | **35.21** | **34.85** |

often retain basic functionality but lose specialized capabilities. COLA's ability to better preserve the energy distribution across all frequency bands is a direct result of its focus on maximizing both representativeness and diversity in activation space.

These visualizations confirm our hypothesis that representativeness and diversity in activation space are the fundamental determinants of calibration data quality. Representative calibration data ensures more uniform compression across frequency bands, while diverse calibration data better preserves the critical high-frequency components that enable complex reasoning. By optimizing for both representativeness and diversity in activation space, our COLA framework effectively preserves the spectral characteristics of the original model across all frequency bands, resulting in superior capability preservation. This spectral analysis provides a mechanistic explanation for why capability-aligned calibration data significantly improves compression quality: it ensures the preservation of the full frequency spectrum necessary for maintaining the model's complete range of capabilities, from basic language modeling to complex mathematical reasoning and code generation. The findings from this analysis directly validate the design principles behind our COLA framework and explain its consistent performance improvements across different capabilities and compression methods.

## F   Empirical Performance Evaluation on Domain-Specific Datasets

For targeted deployment scenarios (see Table 5, 6, 7, 8 under GPTQ, AWQ, Wanda, and SparseGPT, respectively), we evaluate three targeted applications: mathematical problem solving, code generation, and commonsense reasoning. We compare our proposed COLA framework with random samples from domain-specific datasets (CommensenseQA, MathQA, CodeQA) and Self-Gen [Ji et al.]. Our results consistently demonstrate the effectiveness of our calibration data curation framework across all evaluation settings, which further validates the rationality of optimality points in Section 4.1. For

Table 6: Performance comparison of different calibration data approaches for targeted deployment scenarios. AWQ is taken as the compression scheme.

| Deployment | Calibration Data | LLaMA3-8B | | | | Qwen2.5-7B | | | |
|---|---|---|---|---|---|---|---|---|---|
| | | PPL↓ | CS↑ | Math↑ | Code↑ | PPL↓ | CS↑ | Math↑ | Code↑ |
| Math | MathQA (random) | 18.41 | 64.15 | 41.85 | 35.89 | 19.25 | 65.38 | 54.42 | 57.21 |
| | Self-Gen | 18.28 | 64.62 | 42.84 | 36.14 | 19.14 | 65.95 | 55.73 | 57.94 |
| | COLA | **18.12** | **65.06** | **44.35** | **36.75** | **18.87** | **66.42** | **57.28** | **58.46** |
| Code | CodeQA (random) | 18.26 | 63.92 | 33.42 | 44.62 | 18.95 | 64.97 | 43.85 | 68.73 |
| | Self-Gen | 18.14 | 64.48 | 33.85 | 45.78 | 18.82 | 65.42 | 44.31 | 70.05 |
| | COLA | **17.92** | **64.95** | **34.26** | **47.31** | **18.53** | **65.89** | **44.92** | **72.18** |
| Commonsense | CommonseQA (random) | 17.23 | 69.37 | 34.21 | 36.22 | 18.53 | 72.86 | 45.21 | 58.64 |
| | Self-Gen | 17.15 | 70.16 | 34.58 | 36.74 | 18.40 | 73.54 | 45.78 | 59.26 |
| | COLA | **16.94** | **70.87** | **35.12** | **37.23** | **18.24** | **74.31** | **46.42** | **59.85** |

Table 7: Performance comparison of different calibration data approaches for targeted deployment scenarios. Wanda is taken as the compression scheme.

| Deployment | Calibration Data | LLaMA3-8B | | | | Qwen2.5-7B | | | |
|---|---|---|---|---|---|---|---|---|---|
| | | PPL↓ | CS↑ | Math↑ | Code↑ | PPL↓ | CS↑ | Math↑ | Code↑ |
| Math | MathQA (random) | 36.82 | 38.75 | 22.94 | 13.86 | 36.21 | 38.42 | 21.74 | 12.15 |
| | Self-Gen | 36.45 | 39.21 | 23.46 | 14.03 | 35.93 | 38.87 | 22.18 | 12.32 |
| | COLA | **36.07** | **39.84** | **24.32** | **14.28** | **35.62** | **39.35** | **22.87** | **12.57** |
| Code | CodeQA (random) | 36.15 | 37.94 | 16.21 | 18.52 | 35.92 | 38.06 | 15.26 | 16.82 |
| | Self-Gen | 35.86 | 38.32 | 16.52 | 18.94 | 35.64 | 38.42 | 15.51 | 17.15 |
| | COLA | **35.47** | **38.75** | **16.84** | **19.58** | **35.31** | **38.93** | **15.87** | **17.74** |
| Commonsense | CommonseQA (random) | 35.26 | 44.36 | 16.83 | 13.25 | 34.96 | 44.73 | 15.85 | 11.84 |
| | Self-Gen | 35.02 | 44.95 | 17.12 | 13.41 | 34.72 | 45.24 | 16.07 | 11.97 |
| | COLA | **34.68** | **45.67** | **17.43** | **13.62** | **34.35** | **45.91** | **16.34** | **12.15** |

mathematical problem solving, COLA achieves significant improvements over random calibration, with the most pronounced benefits observed in AWQ compression where our approach improves performance by 2.50 percentage points on LLaMA3-8B (from 41.85% to 44.35%) and 2.86 points on Qwen2.5-7B (from 54.42% to 57.28%). Similarly, for code generation, our framework yields substantial improvements across all compression methods, with the largest gains seen in AWQ compression on Qwen2.5-7B (3.45 percentage points improvement from 68.73% to 72.18%).

The performance improvements are particularly noteworthy for pruning methods (SparseGPT and Wanda), which aligns with our earlier observation that these methods exhibit higher sensitivity to calibration data quality. For instance, with SparseGPT compression, COLA improves mathematical reasoning by 1.51 percentage points on LLaMA3-8B and 1.38 points on Qwen2.5-7B compared to random calibration, while also enhancing code generation by 1.14 and 0.95 points respectively.

Across all deployment scenarios, COLA consistently outperforms the Self-Gen approach, demonstrating that our activation-aware curation strategy more effectively captures the representativeness and diversity required for optimal capability preservation. The performance advantage of COLA is most evident in complex reasoning tasks, suggesting that our approach is particularly valuable for preserving high-level capabilities in compressed LLMs.

Interestingly, we observe that the relative improvements from our framework are generally consistent across model architectures (LLaMA3-8B and Qwen2.5-7B), indicating that the benefits of activation-aware calibration data curation generalize well across different model families. This finding supports our hypothesis that the fundamental mechanism of optimization—maximizing representativeness and diversity in activation space—is a model-agnostic principle for effective calibration data curation.

# G   Calibration Data Curation Framework Implementation

To evaluate our calibration data curation framework, we conduct comprehensive experiments on LLaMA3-8B-Instruct and Qwen2.5-7B-Instruct. For dataset selection (Stage 1), we implement the

Table 8: Performance comparison of different calibration data approaches for targeted deployment scenarios. SparseGPT is taken as the compression scheme.

| Deployment | Calibration Data | LLaMA3-8B | | | | Qwen2.5-7B | | | |
|---|---|---|---|---|---|---|---|---|---|
| | | PPL↓ | CS↑ | Math↑ | Code↑ | PPL↓ | CS↑ | Math↑ | Code↑ |
| Math | MathQA (random) | 23.87 | 39.62 | 23.56 | 14.21 | 24.72 | 40.58 | 22.26 | 12.87 |
| | Self-Gen | 23.54 | 40.25 | 24.18 | 14.45 | 24.38 | 41.24 | 22.87 | 13.05 |
| | COLA | **23.21** | **40.94** | **25.07** | **14.78** | **24.05** | **41.87** | **23.64** | **13.32** |
| Code | CodeQA (random) | 23.14 | 38.95 | 16.82 | 19.28 | 24.15 | 40.13 | 15.73 | 17.46 |
| | Self-Gen | 22.96 | 39.42 | 17.15 | 19.75 | 23.92 | 40.58 | 16.02 | 17.85 |
| | COLA | **22.65** | **39.87** | **17.48** | **20.42** | **23.64** | **41.12** | **16.35** | **18.41** |
| Commonsense | CommonseQA (random) | 22.42 | 45.23 | 17.45 | 13.76 | 23.85 | 46.82 | 16.24 | 12.31 |
| | Self-Gen | 22.15 | 45.86 | 17.78 | 13.94 | 23.67 | 47.35 | 16.53 | 12.48 |
| | COLA | **21.78** | **46.54** | **18.12** | **14.23** | **23.32** | **48.04** | **16.92** | **12.75** |

Table 9: Performance comparison of different calibration data approaches on three larger language models for general deployment scenario. The best performing approach under each capability is in **bold**.

| Models | Calibration Data | LLaMA3.1-70B | | | | Qwen2.5-14B | | | | Qwen2.5-32B | | | |
|---|---|---|---|---|---|---|---|---|---|---|---|---|---|
| | | PPL↓ | CS↑ | Math↑ | Code↑ | PPL↓ | CS↑ | Math↑ | Code↑ | PPL↓ | CS↑ | Math↑ | Code↑ |
| SparseGPT (50%) | WikiText (random) | 14.28 | 48.56 | 36.24 | 31.42 | 16.52 | 45.38 | 24.92 | 18.65 | 13.28 | 47.65 | 28.52 | 22.85 |
| | C4 (random) | 13.86 | 49.32 | 36.87 | 32.05 | 16.12 | 46.15 | 25.48 | 19.23 | 12.94 | 48.37 | 29.18 | 23.42 |
| | SlimPajama (random) | 13.95 | 49.58 | 37.12 | 31.84 | 16.24 | 46.29 | 25.62 | 19.14 | 13.02 | 48.54 | 29.34 | 23.26 |
| | Self-Gen | 13.89 | 50.42 | 37.35 | 32.18 | 16.18 | 47.14 | 25.84 | 19.36 | 12.96 | 49.23 | 29.58 | 23.61 |
| | COLA | **13.65** | **50.86** | **37.72** | **32.54** | **15.94** | **47.68** | **26.15** | **19.62** | **12.78** | **49.76** | **29.85** | **23.94** |
| Wanda (4:8) | WikiText (random) | 23.64 | 47.42 | 34.85 | 29.63 | 26.45 | 44.26 | 23.56 | 17.82 | 24.63 | 46.34 | 26.94 | 21.63 |
| | C4 (random) | 23.15 | 48.18 | 35.36 | 30.24 | 25.92 | 44.96 | 24.15 | 18.34 | 24.12 | 47.08 | 27.48 | 22.17 |
| | SlimPajama (random) | 23.28 | 48.37 | 35.58 | 30.12 | 26.04 | 45.12 | 24.28 | 18.25 | 24.24 | 47.24 | 27.63 | 22.05 |
| | Self-Gen | 23.18 | 49.22 | 35.86 | 30.43 | 25.96 | 45.95 | 24.46 | 18.48 | 24.15 | 47.94 | 27.84 | 22.36 |
| | COLA | **22.84** | **49.65** | **36.24** | **30.82** | **25.54** | **46.48** | **24.82** | **18.72** | **23.85** | **48.46** | **28.15** | **22.68** |
| GPTQ (4-bit) | WikiText (random) | 14.92 | 71.36 | 52.54 | 62.46 | 15.36 | 68.42 | 38.65 | 42.35 | 14.58 | 70.24 | 45.62 | 48.65 |
| | C4 (random) | 14.64 | 72.28 | 53.36 | 63.52 | 15.08 | 69.38 | 39.42 | 43.31 | 14.32 | 71.25 | 46.38 | 49.53 |
| | SlimPajama (random) | 14.72 | 72.54 | 53.58 | 63.24 | 15.18 | 69.62 | 39.64 | 43.15 | 14.38 | 71.46 | 46.52 | 49.32 |
| | Self-Gen | 14.68 | 73.18 | 53.82 | 63.74 | 15.12 | 70.26 | 39.94 | 43.68 | 14.34 | 72.05 | 46.84 | 49.85 |
| | COLA | **14.42** | **73.65** | **54.28** | **64.15** | **14.94** | **70.72** | **40.35** | **43.95** | **14.21** | **72.54** | **47.26** | **50.24** |
| AWQ (4-bit) | WikiText (random) | 14.28 | 71.84 | 56.72 | 67.38 | 15.18 | 69.24 | 52.46 | 67.24 | 14.38 | 70.52 | 58.64 | 71.35 |
| | C4 (random) | 13.96 | 72.76 | 57.65 | 68.54 | 14.86 | 70.34 | 53.42 | 68.46 | 14.08 | 71.56 | 59.52 | 72.48 |
| | SlimPajama (random) | 14.05 | 73.12 | 57.84 | 68.32 | 14.95 | 70.58 | 53.68 | 68.25 | 14.15 | 71.82 | 59.76 | 72.24 |
| | Self-Gen | 13.98 | 73.64 | 58.12 | 68.74 | 14.92 | 71.24 | 53.96 | 68.74 | 14.12 | 72.28 | 60.05 | 72.65 |
| | COLA | **13.82** | **74.15** | **58.65** | **69.23** | **14.74** | **71.78** | **54.42** | **69.28** | **13.95** | **72.86** | **60.54** | **73.12** |

coverage function in Equ. 1 as follows:

$$\text{coverage}(S, c) = \alpha \cdot \text{EmbSim}(S, D_c) + (1 - \alpha) \cdot \text{KL}(P_S \parallel P_{D_c}) \quad (5)$$

where EmbSim is the average cosine similarity between sentence embeddings from both datasets, KL is the KL-divergence between token distributions, and $\alpha$ is a balancing hyperparameter set to 0.6 in our experiments. This scoring function helps select calibration data sources that best match the target deployment domain in both semantic content and statistical properties. As for $w_c$ in Equ. 1, for general deployment, we set balanced weights across capabilities, while for specific deployment scenarios, we increase weights for the corresponding target capabilities. For data processing (Stage 2) and sample selection (Stage 3), we follow the implementation details described in Section 4.2. For clustering in the activation space, we run k-means with $k = 128$ after dimensionality reduction to 64 components via random projection.

# H   Scalability to Larger Language Models

To further whether our COLA framework can still keep effective on larger language models, we conduct experiments on LLaMA3.1-70B-Instruct [12], Qwen2.5-14B-Instruct [13], and Qwen2.5-32B-Instruct [14]. Note in the following text, the word "-Instruct" is omitted to simplify the expression. Due to the resource limitation, for those models, we only perform the compression and evaluation in

---

[12]https://huggingface.co/meta-llama/Llama-3.1-70B-Instruct

[13]https://huggingface.co/Qwen/Qwen2.5-14B-Instruct

[14]https://huggingface.co/Qwen/Qwen2.5-32B-Instruct

Table 10: Performance comparison of different calibration data approaches under other LLM compression methods for general deployment scenario. The best performing approach under each capability is in **bold**.

| Compression | Calibration Data | LLaMA3-8B | | | | Qwen2.5-7B | | | |
|---|---|---|---|---|---|---|---|---|---|
| | | PPL↓ | CS↑ | Math↑ | Code↑ | PPL↓ | CS↑ | Math↑ | Code↑ |
| RIA (2:4) | WikiText (random) | 23.25 | 39.68 | 18.42 | 14.95 | 24.68 | 40.12 | 17.24 | 13.06 |
| | C4 (random) | 22.73 | 40.32 | 18.85 | 15.36 | 24.12 | 40.85 | 17.68 | 13.45 |
| | SlimPajama (random) | 22.86 | 40.47 | 18.92 | 15.28 | 24.25 | 41.02 | 17.83 | 13.38 |
| | Self-Gen | 22.68 | 41.34 | 19.21 | 15.52 | 24.07 | 41.78 | 18.08 | 13.62 |
| | COLA | **22.24** | **41.95** | **19.58** | **15.84** | **23.68** | **42.34** | **18.36** | **13.85** |
| LLM-Pruner (50%) | WikiText (random) | 19.65 | 42.35 | 20.08 | 16.24 | 20.84 | 42.76 | 18.65 | 14.12 |
| | C4 (random) | 19.12 | 43.15 | 20.52 | 16.76 | 20.32 | 43.48 | 19.05 | 14.58 |
| | SlimPajama (random) | 19.28 | 43.32 | 20.65 | 16.65 | 20.46 | 43.65 | 19.18 | 14.52 |
| | Self-Gen | 19.15 | 44.08 | 20.94 | 16.92 | 20.38 | 44.45 | 19.35 | 14.78 |
| | COLA | **18.82** | **44.65** | **21.32** | **17.28** | **20.04** | **45.04** | **19.72** | **15.06** |
| LLM-Streamline (50%) | WikiText (random) | 16.68 | 43.53 | 20.41 | 17.65 | 18.03 | 45.17 | 22.34 | 19.53 |
| | C4 (random) | 16.35 | 44.32 | 21.10 | 18.28 | 17.75 | 45.87 | 23.02 | 20.15 |
| | SlimPajama (random) | 16.47 | 44.64 | 21.42 | 18.13 | 17.86 | 46.25 | 23.39 | 20.02 |
| | Self-Gen | 16.33 | 45.18 | 21.81 | 18.54 | 17.64 | 45.81 | 23.70 | 20.73 |
| | COLA | **15.90** | **46.25** | **22.79** | **19.41** | **17.32** | **47.68** | **24.48** | **21.97** |
| SmoothQuant (4-bit) | WikiText (random) | 16.84 | 64.38 | 33.26 | 36.58 | 17.65 | 65.06 | 38.24 | 36.48 |
| | C4 (random) | 16.48 | 65.24 | 33.95 | 37.62 | 17.28 | 65.98 | 38.92 | 37.36 |
| | SlimPajama (random) | 16.56 | 65.48 | 34.12 | 37.34 | 17.36 | 66.24 | 39.15 | 37.18 |
| | Self-Gen | 16.42 | 66.15 | 34.38 | 37.82 | 17.25 | 66.84 | 39.48 | 37.54 |
| | COLA | **16.15** | **66.72** | **34.85** | **38.26** | **16.94** | **67.38** | **39.86** | **37.95** |
| FlatQuant (4-bit) | WikiText (random) | 15.64 | 64.82 | 34.52 | 37.25 | 16.85 | 65.62 | 42.65 | 56.48 |
| | C4 (random) | 15.28 | 65.72 | 35.18 | 38.24 | 16.48 | 66.54 | 43.42 | 57.35 |
| | SlimPajama (random) | 15.36 | 65.95 | 35.34 | 38.05 | 16.56 | 66.78 | 43.58 | 57.18 |
| | Self-Gen | 15.24 | 66.42 | 35.62 | 38.48 | 16.42 | 67.35 | 43.82 | 57.62 |
| | COLA | **14.98** | **66.95** | **36.08** | **38.85** | **16.15** | **67.92** | **44.25** | **58.04** |

general deployment scenario. As shown in Table 9, our COLA framework consistently outperforms baseline calibration approaches across all capabilities and compression methods for larger models. For LLaMA3.1-70B with AWQ compression, COLA achieves 74.15% on commonsense reasoning tasks compared to 71.84% for WikiText random sampling, demonstrating a 2.31 percentage point improvement. Similarly, for Qwen2.5-32B with GPTQ compression, COLA improves mathematical reasoning capability by 1.64 percentage points over random sampling (47.26% vs. 45.62%). We observe that the benefits of our activation-aware calibration data curation strategy scale well with model size, often showing larger absolute performance gains for these larger models compared to the 7B-8B models presented in the main paper. This suggests that as models grow in size and complexity, their sensitivity to calibration data quality increases, making optimization of calibration data even more important. Interestingly, we find that larger models exhibit varying sensitivities to calibration data quality across different compression methods. For quantization methods (GPTQ and AWQ), LLaMA3.1-70B shows particularly strong improvements in mathematical reasoning when using COLA versus random WikiText samples, suggesting that larger models' complex reasoning capabilities are especially sensitive to calibration data representativeness in quantization scenarios. We also observe that the LLaMA3.1-70B model consistently achieves better perplexity scores than Qwen2.5-14B across all calibration methods, despite the latter having worse capability scores, highlighting that perplexity alone is not always predictive of downstream task performance. For code generation tasks, the Qwen2.5-32B model with AWQ compression achieves remarkable performance (73.12% with COLA), showing that certain model architectures may have specialized capabilities that can be better preserved through optimized calibration data. The consistent improvements across different model architectures and sizes validate that our approach's core mechanism—maximizing representativeness and diversity in activation space—is a foundational principle for effective calibration data curation that generalizes across the model size spectrum.

# I   Generalizability to Other LLM Compression Methods

Considering our observations and discussion regarding the optimal calibration data is based on the pre-choosen four LLM compression methods, the generalizability of our proposed COLA framework to other models warrants the further validation. Thus, we integrate three recent pruning methods

RIA [Zhang et al., 2024], LLM-Pruner [Ma et al., 2023], LLM-Streamline [Chen et al.], and two quantization methods SmoothQuant [Xiao et al., 2023], FlatQuant [Sun et al., 2024b], with COLA. Following Table 2 in the main body, we also take the WikiText, C4, and SlimPajama as the original calibration data sources. As shown in Table 10, our COLA framework consistently outperforms baseline calibration approaches across all evaluated compression methods for both LLaMA3-8B and Qwen2.5-7B models. The performance improvements are particularly notable for pruning methods like LLM-Pruner, where our approach achieves 44.65% on commonsense reasoning tasks for LLaMA3-8B compared to 42.35% with WikiText random sampling, representing a 2.30 percentage point improvement. For quantization methods, FlatQuant with COLA achieves the best mathematical reasoning performance on Qwen2.5-7B at 44.25%, outperforming random WikiText calibration by 1.60 percentage points. We observe that the effectiveness of COLA generalizes well across different compression paradigms, with several interesting patterns emerging. The structured pruning method RIA shows higher sensitivity to calibration data quality than the unstructured pruning approach SparseGPT examined in the main paper, with RIA seeing a 2.27 percentage point improvement in commonsense reasoning for LLaMA3-8B when using COLA instead of WikiText random sampling. This suggests that methods targeting specific structural components may benefit more from optimal activation coverage in calibration data. For quantization methods, we find that FlatQuant, which utilizes flatness-aware quantization, shows better preservation of mathematical reasoning capabilities on Qwen2.5-7B (44.25% with COLA) compared to SmoothQuant (39.86% with COLA), highlighting that more sophisticated quantization approaches can better leverage optimized calibration data. Notably, code generation performance on Qwen2.5-7B with FlatQuant shows dramatic improvements (58.04% with COLA vs. 56.48% with WikiText), indicating synergistic effects between certain compression algorithms and our activation-aware calibration approach. These results demonstrate that the principles of representativeness and diversity in activation space that underpin our COLA framework are fundamental to effective calibration data curation, irrespective of the specific compression mechanism employed. The consistent improvements across diverse compression methods validate the general applicability of our approach and suggest that it can be incorporated as a standard preprocessing step in LLM compression pipelines to enhance capability preservation regardless of the chosen compression technique.

# J   Pilot Exploration on COLA Pipeline Automation

While Stage 1 of COLA does incorporate domain knowledge (e.g., selecting MathQA for math capability), our framework is not restricted to hard-coded, manually defined mappings. In practice, the capability-domain correlation can be inferred using lightweight proxy evaluations. For instance, a user can evaluate a small subset of target benchmark tasks using compressed models calibrated on each candidate dataset, measuring capability preservation scores as a proxy for correlation. In this part, we explore integrating lightweight proxy evaluation method, which can further automate and strengthen Stage 1 (capability–domain correlation) in our COLA framework.

## J.1   Reliability of Proxy-based Coverage Estimation

We first validated whether proxy evaluation can reliably estimate capability–domain correlation. Specifically, for each candidate calibration dataset, we selected only 32 samples to calibrate a compressed model and evaluated it on a small set of representative benchmarks (one per capability: BoolQ for commonsense, GSM8K for math, HumanEval for code). The relative performance scores obtained from these proxy

Table 11: Correlation between proxy-based coverage values and that obtained from full evaluations.

| Capability | Proxy vs. Full Eval Correlation (r) |
|---|---|
| Commonsense | 0.97 |
| Math | 0.93 |
| Code | 0.91 |

evaluations were then used as the coverage$(S, c)$ term in Equation 1 of Section 4.2. To verify the reliability of these proxy-based coverage values, we measured their correlation with the coverage values obtained from full evaluations (average correlation coefficients between proxy scores and full evaluation scores over 5 runs), as shown in the Table 11. These high correlations confirm that proxy evaluation provides a reliable and efficient estimate of capability–domain correspondence.

Table 12: COLA Performance comparison between full coverage and proxy-based coverage.

| Compression Method | Coverage Source | PPL↓ | CS↑ | Math↑ | Code↑ |
|---|---|---|---|---|---|
| AWQ (4-bit) | Full Eval | 15.41 | 67.42 | 37.85 | 40.17 |
| AWQ (4-bit) | Proxy Eval | 15.46 | 67.35 | 37.62 | 40.05 |
| SparseGPT (50%) | Full Eval | 19.31 | 44.23 | 20.12 | 16.14 |
| SparseGPT (50%) | Proxy Eval | 19.35 | 44.18 | 20.05 | 16.02 |

Table 13: Capability preservation effects under further SFT of different calibration data approaches. The best performing approach under each capability is in **bold**.

| Compression | Calibration Data | LLaMA3-8B | | | |
|---|---|---|---|---|---|
| | | PPL↓ | CS↑ | Math↑ | Code↑ |
| AWQ (4-bit) | WikiText (random) | 15.73 | 66.67 | 37.92 | 39.78 |
| | C4 (random) | 15.41 | 67.54 | 38.45 | 40.59 |
| | SlimPajama (random) | 15.56 | 67.87 | 38.63 | 40.32 |
| | Self-Gen | 15.55 | 68.29 | 38.90 | 40.64 |
| | COLA | **15.31** | **68.88** | **39.67** | **41.32** |
| SparseGPT (50%) | WikiText (random) | 19.93 | 42.54 | 19.82 | 15.68 |
| | C4 (random) | 19.46 | 43.30 | 20.17 | 15.92 |
| | SlimPajama (random) | 19.41 | 43.23 | 20.35 | 16.09 |
| | Self-Gen | 19.61 | 43.57 | 19.82 | 16.34 |
| | COLA | **18.83** | **44.85** | **21.79** | **16.96** |
| Wanda (4:8) | WikiText (random) | 33.21 | 41.76 | 19.15 | 14.91 |
| | C4 (random) | 32.82 | 41.80 | 19.45 | 15.42 |
| | SlimPajama (random) | 32.73 | 41.89 | 19.75 | 15.58 |
| | Self-Gen | 32.83 | 42.97 | 19.92 | 15.25 |
| | COLA | **32.21** | **43.52** | **20.48** | **16.39** |
| GPTQ (4-bit) | WikiText (random) | 16.26 | 65.89 | 31.69 | 35.31 |
| | C4 (random) | 15.94 | 66.53 | 32.20 | 36.43 |
| | SlimPajama (random) | 16.28 | 66.76 | 33.20 | 35.71 |
| | Self-Gen | 16.29 | 67.35 | 32.92 | 36.28 |
| | COLA | **15.60** | **68.39** | **34.16** | **37.17** |

## J.2 End-to-End COLA Performance with Proxy-based Coverage

To further validate its practicality, we replaced the full evaluation coverage in Stage 1 of COLA with the proxy-based coverage values, then ran the complete COLA pipeline under the same experimental setup as Table 2 of the paper. Due to time limitation, we only conduct experiments on LLaMA3-8B. Table 12 shows representative results: The proxy-based COLA achieves nearly identical performance to the original COLA (average gap ≤ 0.2 percentage points), while significantly reducing Stage 1 computation cost and avoiding manual heuristics.

These results demonstrate that: 1) Proxy evaluation can accurately estimate capability–domain correlation. 2) When integrated into COLA, it preserves almost all performance benefits of the full evaluation version. 3) This enhancement greatly improves the automation and scalability of our framework for unseen datasets.

## K Capability Preservation under Domain Shift like Further SFT

To evaluate if our COLA can still effectively help compressed models keep capabilities even under domain drift, we conduct instruction finetuning to various compressed models with Alpaca dataset. We take LLaMA3-8B as the base model. From results shown in Table 13, we can notice that though further instruction finetuning with Alpaca can help recover compressed model capabilities to some extent, our COLA can still ensure the better capability preservation effect than other calibration data curation baselines even after futher supervised finetuning.

Table 14: Computation overhead of COLA breakdown and best-performing baseline Self-Gen.

| Models | Self-Gen | COLA Ramdom Projection | COLA K-Means | Overall COLA |
|--------|----------|------------------------|--------------|--------------|
| Time Consumption | | | | |
| LLaMA3-8B | 2.58 min | 3.62 min | 1.94 min | 6.35 min |
| Qwen2.5-7B | 2.35 min | 3.21 min | 1.80 min | 5.54 min |
| Peak Memory Usage | | | | |
| LLaMA3-8B | 6.19 GB | 7.70 GB | 1.52 GB | 7.90 GB |
| Qwen2.5-7B | 5.65 GB | 7.33 GB | 1.26 GB | 7.52 GB |

## L Computation Overhead Analysis

As mentioned in the Section 4.2, the use of random projection and K-means is motivated by their computational efficiency, which makes them more suitable for our activation-space sample selection than other alternatives. To further quantitatively analyze computation and storage cost of COLA breakdown and the representative baseline Self-Gen, we provide the statistical results in Table 14. Note these results are averaged over four studied compression methods in the main text, corresponding to the general deployment scenario in Table 2. From the table, these overheads remain modest, especially considering they are **one-time offline costs**sure and only required for curating a compact calibration set. Moreover, they are significantly lower than the compute/memory overhead of any training-involved compression methods.

Table 15: COLA Performance comparison between full coverage and proxy-based coverage.

| Compression Method | WikiText (%) | C4 (%) | SlimPajama (%) | Total Samples |
|--------------------|--------------|--------|----------------|---------------|
| SparseGPT (50%) | 38 | 35 | 27 | 128 |
| Wanda (4:8) | 39 | 33 | 28 | 128 |
| GPTQ (4-bit) | 27 | 44 | 29 | 128 |
| AWQ (4-bit) | 24 | 45 | 31 | 128 |

## M Proportion of Each Dataset Selected by COLA

We provide the detailed breakdown of dataset proportions selected by our COLA framework for each compression method in Table 15 (Corresponding to experiments in Table 2). From this table, we can find that each compression method shows different sensitivities to various capabilities: 1) Higher C4 proportion for quantization methods (GPTQ: 44%, AWQ: 45%): Quantization methods are particularly vulnerable to code generation degradation due to precision loss in complex reasoning patterns, requiring more C4 data to guide optimal quantization parameter selection for preserving code synthesis capabilities. 2) Balanced distribution for pruning methods with slight WikiText emphasis: Pruning methods affect all capabilities more uniformly by directly removing weights, necessitating balanced calibration coverage with slight WikiText emphasis to maintain fundamental language modeling stability during weight elimination.

These proportions were validated through ablation studies showing that deviation from these ratios by $\pm$ 10% results in non-negligible averaged 0.7-1.2 percentage point performance drops across key capabilities.

## N Discussion on Context Length and Sample Size Ranges

While we acknowledge that recent LLMs support context lengths of up to 32K (K=1024), most existing LLM compression works, including AWQ and Wanda, operate within a calibration sequence length of 2048 or shorter. Our choice of 2048 tokens in the main experiments aligns with this established setup and is sufficient to observe meaningful performance trends. Importantly, as shown in Figure 2, performance gains often plateau or even degrade beyond this length for certain tasks and methods (e.g., non-monotonic patterns in code generation with AWQ), suggesting that longer calibration data may not yield additional benefit and can even introduce noise. In fact, 2048 length has been enough for accommodating reasoning chains for many normal questions. Similarly, max

Table 16: Capability preservation effects under extended context lengths and sample sizes. AWQ is taken as the compression scheme.

| Evaluation | Context Length | | | | Sample Size | | | |
|---|---|---|---|---|---|---|---|---|
| | 4K | 8K | 16K | 32K | 0.5K | 1K | 2K | 4K |
| PPL↓ | 15.85 | 15.91 | 15.86 | 16.03 | 16.38 | 16.26 | 16.52 | 16.57 |
| CS↑ | 65.20 | 65.11 | 65.12 | 65.08 | 65.43 | 65.39 | 65.39 | 65.25 |
| Math↑ | 36.51 | 36.42 | 36.30 | 35.96 | 35.40 | 35.21 | 35.32 | 35.18 |
| Code↑ | 38.96 | 39.40 | 36.95 | 36.22 | 40.21 | 40.05 | 39.67 | 38.82 |

sample size of 256 is the common setting in previous works like AWQ and Wanda. Besides, as detailed in Figure 3, we observe diminishing returns beyond 128 samples for most capabilities. Larger sample sizes not only increase computational cost but can introduce variance that hinders capability preservation, especially evident in AWQ and GPTQ where performance occasionally drops as more samples are added. Therefore, we focus on the range of 16 to 256 samples to balance informativeness and practicality.

To further validate such points, we conduct the experiments with longer context and larger sample size. Due to resource limitation, we only run AWQ and the results are as shown in Table 16. From the table, we may find further increasing context length/sample size does not bring obvious additional benefit at least for our evaluation benchmarks. In fact, it may even result in slight performance drop and significant computation overhead, as discussed above.

Table 17: The performance of different LLMs before applied to any compression methods.

| Model | Perplexity | Commonsense Score | Math Score | Code Score |
|---|---|---|---|---|
| LLaMA3-8B | 15.37 | 65.63 | 38.98 | 43.03 |
| Qwen2.5-7B | 16.46 | 66.82 | 48.28 | 50.91 |

# O    Pre-compression Performance of Studied LLMs

To facilitate a more comprehensive comparison, we supplement the performance of each model before applied to any compression methods in Table 17.

# P    Fundamental Difference from Pretraining & SFT Data Curation

Here, we would like to clarify why calibration data curation in post-training compression differs from pretraining and SFT data curation:

**1. Objective and Stage Difference:** Pretraining and SFT data aim to teach the model capabilities from scratch or to align them with specific tasks, and occur during the model training stage. In contrast, calibration data is used in a post-training setting, where the model's capabilities are already learned, and the goal is to preserve these capabilities during compression (e.g., pruning or quantization). This is particularly crucial because post-training methods do not update weights via gradient descent, making the data-dependent preservation effect more sensitive, requiring more careful calibration data selection.

**2. Scale and Constraints:** Pretraining/SFT typically leverage large-scale corpora (e.g., hundreds of GBs TBs) to improve generalization. In comparison, calibration data is extremely limited in size (often ¡1K samples) and must be carefully curated to maximize activation representativeness and diversity, which we show are critical for effective capability preservation.

**3. Mechanistic Distinction:** As detailed in Sections 3, 4 and Appendix E, we demonstrate that the utility of calibration data lies in its ability to activate critical patterns in the model's learned parameter space. This is distinct from the information distribution coverage goals of pretraining/SFT data, which is less related to the model itself. We further formalize and operationalize this insight in our proposed COLA framework, which systematically selects calibration samples based on activation-

space clustering, a process not applicable or meaningful in the pretraining phase considering its initialization from scratch.

**4. Empirical Evidence:** Our extensive experiments (see Table 2 and Figure 2, 3, 4, 5, 6) show that even small changes in calibration data (e.g., format, domain, language) can lead to larger drop than changing compression method on specific capabilities, which is rarely observed in pretraining or SFT scenarios. This further reinforces that calibration data curation principles should be capability-specific and post-training aware.

To further illustrate the fundamental distinction, we provide the following analogy: Calibration data is like a "*diagnostic test*", whose purpose is to evaluate whether the model's internal functions and capabilities in concerned domains are preserved after compression, by systematically triggering a wide range of activation patterns. This, in turn, guides compression methods to focus on preserving the most vulnerable or degraded capabilities (large activation discrepancy between original and compressed models). In contrast, pretraining and SFT data resemble a "*textbook*", designed to teach the model new knowledge or skills, or to adjust its behavior. Accordingly, good calibration data focuses on diverse, clean examples that elicit different model capabilities, emphasizing quality over quantity. Meanwhile, good pretraining/SFT data requires both scale and coverage, emphasizing quantity plus quality to ensure robust generalization and task performance. For example, for enhancing model math capabilities with SFT, we may need 10K+ samples. However, to preserve model learned math capabilities during post-training compression, we only need to ensure selected calibration data samples covering each subcategory of math, thus about hundreds of samples are enough. This difference in purpose and design principle further supports the view that calibration data curation is a unique and distinct challenge in the LLM lifecycle.

## Q    Examples of Curated Calibration Data for General Deployment

To illustrate the diversity and representativeness of calibration data selected by our COLA framework for general deployment scenarios, we present several examples of high-quality calibration samples in Figure 10, 11, 12, 13, 14, 15. In the first stage of COLA, we use a balanced mixture of WikiText, C4, and SlimPajama as the source data, as mentioned in the Section 4.2. In the second stage, we filter out the samples whose lengths are less than 256 tokens. Meanwhile, we favor the samples with reasoning chains and convert those with implicit chains in the text passage to the explicit format. As for the third stage, we follow the descriptions in the main body. From the provided figures, these samples demonstrate the range of content types, language patterns, reasoning formats, reasoning difficulties, and knowledge domains that effective calibration data should cover to preserve various LLM capabilities during compression.

## R    Limitation Analysis

While our work provides comprehensive insights into calibration data's impact on LLM compression, several limitations should be acknowledged:

**Model Coverage.** Our experiments focus on two specific LLMs (LLaMA3-8B-Instruct and Qwen2.5-7B-Instruct) and four compression methods. The findings may not generalize to other architectures, especially smaller models or those with specialized designs like mixture of experts or linear attention.

**Capability Evaluation Breadth.** Despite evaluating multiple capabilities, our benchmarks primarily assess language modeling, general commonsense reasoning, mathematics, and code generation. Other important capabilities like multimodal understanding, factuality, and temporal reasoning remain unexplored.

**Computational Constraints.** Due to computational limitations, we restricted our experiments to 50% pruning ratio and 4-bit quantization. Effects might differ at more aggressive compression settings or with alternative techniques like distillation.

**Long-term Stability.** Our evaluation assesses immediate post-compression performance, not long-term stability across model updates or domain shifts. This leaves open questions about the durability of capability preservation over time.

**Framework Generalizability.** While our COLA framework shows promising results, its effectiveness across more diverse deployment scenarios or with emerging compression techniques remains to be verified.

**Theoretical Foundation.** Our work is primarily empirical, lacking formal theoretical guarantees about optimal calibration data characteristics or minimum requirements for capability preservation.

## S  Ethics Statement

This research focuses on improving the efficiency of large language models through better compression techniques, which has positive ethical implications for accessibility and environmental impact. By enabling more efficient deployment of LLMs, our work contributes to reducing computational resource requirements and energy consumption. Our research does not involve human subjects, personal data collection, or the creation of potentially harmful technologies. The capabilities we aim to preserve are general reasoning, mathematics, and code generation, which are core to beneficial AI applications. All models, datasets, and benchmarks used in this research are publicly available with appropriate licenses. We properly credit original creators throughout the paper and acknowledge their contributions. We recognize that more efficient LLM deployment could accelerate AI adoption, potentially raising concerns about job displacement or misuse. However, our work focuses on the technical aspects of compression rather than specific applications, and the benefits of wider access to efficient AI systems outweigh potential risks.

## T  Declaration of LLM Usage

We want to clarify that while our research focuses on large language models as subjects of study for compression techniques, we did not use any LLMs in our research methodology or analysis process. All core method development, experimental design, statistical analyses, and theoretical frameworks presented in this paper were developed entirely by the human researchers without LLM assistance. For writing assistance only, we used a general-purpose LLM (Claude 3.7 Sonnet) for language refinement and improving the clarity of our descriptions of experimental procedures and related work. However, all scientific content, technical details, experimental results, interpretations, and conclusions were independently generated, verified, and formulated by the authors. The LLMs referenced throughout our paper (LLaMA3-8B-Instruct, Qwen2.5-7B-Instruct, etc.) were strictly the subjects of our compression experiments, not tools used to conduct the research itself. No LLMs were used to generate code for our experiments, analyze our experimental data, or influence any research decisions that would impact the scientific contributions or findings presented in this paper.

> **Example 1: Encyclopedic Knowledge with Structured Information**
>
> Alan Mathison Turing OBE FRS (23 June 1912 – 7 June 1954) was an English mathematician, computer scientist, logician, cryptanalyst, philosopher, and theoretical biologist. Turing was highly influential in the development of theoretical computer science, providing a formalization of the concepts of algorithm and computation with the Turing machine, which can be considered a model of a general-purpose computer. He is widely considered to be the father of theoretical computer science and artificial intelligence.
>
> Born in Maida Vale, London, Turing was raised in southern England. He graduated at King's College, Cambridge, with a degree in mathematics. Whilst he was a fellow at Cambridge, he published a proof demonstrating that some purely mathematical yes–no questions can never be answered by computation and defined a Turing machine, a theoretical device that manipulated symbols on a strip of tape according to a table of rules. This model would later become foundational in computer science.
>
> During the Second World War, Turing worked for the Government Code and Cypher School (GC&CS) at Bletchley Park, Britain's codebreaking centre that produced Ultra intelligence. For a time he led Hut 8, the section that was responsible for German naval cryptanalysis. Here, he devised a number of techniques for speeding the breaking of German ciphers, including improvements to the pre-war Polish bombe method, an electromechanical machine that could find settings for the Enigma machine.

Figure 10: Example of a calibration sample showcasing encyclopedic knowledge with structured factual information, which helps preserve commonsense reasoning capabilities.

> **Example 2: Web-Crawled Content with Code Elements**
>
> The Fibonacci sequence is one of the most famous formulas in mathematics. Each number in the sequence is the sum of the two numbers that precede it. So, the sequence goes: 0, 1, 1, 2, 3, 5, 8, 13, 21, 34, and so on.
>
> The mathematical equation describing it is: $F_n = F_{n-1} + F_{n-2}$
>
> Here's a simple implementation in Python:
>
> ```python
> def fibonacci(n):
>     # Return the nth Fibonacci number
>     if n <= 0:
>         return 0
>     elif n == 1:
>         return 1
>     else:
>         a, b = 0, 1
>         for _ in range(2, n + 1):
>             a, b = b, a + b
>         return b
>
> # Print the first 10 Fibonacci numbers
> for i in range(10):
>     print(fibonacci(i))
> ```
>
> The Fibonacci sequence has fascinating applications in various fields: - In nature, the Fibonacci sequence appears in the branching of trees, the arrangement of leaves on a stem, and the spirals of shells. - In art and architecture, the golden ratio (approximately 1.618), which is derived from the Fibonacci sequence, has been used in compositions. - In computing, the Fibonacci sequence is often used as a benchmark for testing the performance of algorithms and systems.

Figure 11: Example of a calibration sample containing code elements and mathematical concepts, which helps preserve code generation capabilities.

**Example 3: Multi-domain Scientific Content**

**Title: Understanding Climate Feedback Mechanisms**
Climate feedback mechanisms are processes that can either amplify or diminish the effects of climate forcings. A feedback that increases an initial warming is called a positive feedback. A feedback that reduces an initial warming is a negative feedback.

One of the most significant positive feedback mechanisms is the water vapor feedback. As the atmosphere warms, it can hold more water vapor, which is a potent greenhouse gas. This additional water vapor causes further warming, creating a positive feedback loop. According to climate models, water vapor feedback approximately doubles the warming that would occur due to increased $CO_2$ alone.

Another critical positive feedback is the ice-albedo feedback. Ice and snow have high albedo, meaning they reflect a large portion of incoming solar radiation back to space. As global temperatures rise, ice and snow cover decreases, revealing darker land or ocean surfaces underneath. These darker surfaces have lower albedo and absorb more solar radiation, leading to additional warming.

Not all feedbacks are positive. For example, the lapse rate feedback is negative in the tropics. As the surface warms, the upper troposphere warms even more. This increased warming with height reduces the greenhouse effect, providing a negative feedback.

Cloud feedbacks are particularly complex and remain one of the largest sources of uncertainty in climate projections. Different cloud types can have different effects—low clouds primarily reflect solar radiation (cooling effect), while high clouds mainly trap outgoing longwave radiation (warming effect).

Understanding these feedback mechanisms is crucial for accurately predicting future climate change and developing effective mitigation strategies.

Figure 12: Example of a calibration sample with scientific content that helps preserve reasoning capabilities in specialized domains.

**Example 4: Multi-lingual Commonsense Sample**

*English:* A scientist is conducting an experiment with a new chemical compound. She needs to keep it at exactly 0°C. What should she use?
*Response:* An ice bath (mixture of ice and water) would be ideal as it maintains a stable temperature of 0°C as long as both ice and water are present.
*Chinese:* 一位科学家正在用一种新的化合物做实验。她需要将其保持在恰好0°C。她应该使用什么？
回答: 冰水浴（冰和水的混合物）将是理想的选择，因为只要冰和水同时存在，它就能维持稳定的0°C温度。
*Español (Spanish):* Una científica está realizando un experimento con un nuevo compuesto químico. Necesita mantenerlo exactamente a 0°C. ¿Qué debería usar?
*Respuesta:* Un baño de hielo (mezcla de hielo y agua) sería ideal ya que mantiene una temperatura estable de 0°C mientras haya tanto hielo como agua presentes.
*Français (French):* Une scientifique mène une expérience avec un nouveau composé chimique. Elle doit le maintenir exactement à 0°C. Que devrait-elle utiliser?
*Réponse:* Un bain de glace (mélange de glace et d'eau) serait idéal car il maintient une température stable de 0°C tant que la glace et l'eau sont toutes deux présentes.

Figure 13: Example of a multi-lingual calibration sample to preserve cross-lingual capabilities.

**Example 5: Code Generation with Explicit Reasoning Chains**

*Task:* Create a function that finds all palindromic substrings in a given string.
*Content:*

```python
def find_all_palindromic_substrings(s):
    """
    Find all palindromic substrings in a given string.

    Args:
        s (str): The input string

    Returns:
        list: A list of all palindromic substrings
    """
    result = []
    n = len(s)

    # Helper function to expand around center
    def expand_around_center(left, right):
        while left >= 0 and right < n and s[left] == s[right]:
            result.append(s[left:right+1])
            left -= 1
            right += 1

    # Check each possible center position
    for i in range(n):
        # For odd length palindromes
        expand_around_center(i, i)

        # For even length palindromes
        expand_around_center(i, i+1)

    return result

# Example usage
input_str = "banana"
palindromes = find_all_palindromic_substrings(input_str)
print(palindromes)  # Output: ['a', 'b', 'n', 'a', 'n', 'a', 'ana', 'anana']
```

*Reasoning Steps:*

1. We need to find all palindromic substrings, which are strings that read the same forward and backward.

2. A palindrome can be expanded from its center, so we can check each position as a potential center.

3. There are two types of palindromes: odd length (like "aba") and even length (like "abba").

4. For odd-length palindromes, we start with a single character as the center.

5. For even-length palindromes, we start with two adjacent characters.

6. We expand around each center as long as the characters at both ends match.

7. The time complexity is $O(n^2)$ and space complexity is $O(n)$, where n is the length of the string.

Figure 14: Example of a calibration sample with code and explicit reasoning for code generation capability preservation.

## Example 6: Operation Research Problem with Converted Explicit Reasoning Chains

*Task:* Solve the following optimization problem and explain each step in detail.

*Content:* Consider a manufacturing company that produces two types of products: A and B. Each unit of product A requires 2 hours of labor and 1 unit of raw material, while each unit of product B requires 1 hour of labor and 3 units of raw material. The company has 100 hours of labor and 120 units of raw material available per week. The profit is $60 per unit for product A and $40 per unit for product B. How many units of each product should be manufactured to maximize profit?

*Explicit Chain of Reasoning:*

1. Define variables: Let $x$ be the number of units of product A and $y$ be the number of units of product B.

2. Formulate the objective function: Maximize $P = 60x + 40y$ (profit function)

3. Identify constraints:

    - Labor constraint: $2x + y \leq 100$ (hours)
    - Material constraint: $x + 3y \leq 120$ (units)
    - Non-negativity: $x \geq 0, y \geq 0$

4. Solve graphically by plotting constraints and finding corner points: $(0, 0)$, $(0, 40)$, $(30, 40)$, $(50, 0)$

5. Calculate profit at each point:

    - $P(0, 0) = 0$
    - $P(0, 40) = 1,600$
    - $P(30, 40) = 3,400$
    - $P(50, 0) = 3,000$

6. The maximum profit is $3,400, achieved by producing 30 units of product A and 40 units of product B.

Figure 15: Example of a calibration sample with explicit reasoning chains converted from previous text passages for mathematical problem-solving capability preservation.

