# OpenReview forum: "Preserving LLM Capabilities through Calibration Data Curation: From Analysis to Optimization"
_NeurIPS.cc/2025/Conference — NeurIPS 2025 poster_

### Official Review · Reviewer_1Hx7 · 2025-07-01

**Clarity:** 4
**Significance:** 3
**Originality:** 4
**Rating:** 4
**Confidence:** 2

**Summary:**

This paper studies how calibration data affects the ability of quantized LLMs to keep their skills. The authors test different data properties on pruning and quantization methods. They propose a three-stage COLA framework to pick the best samples by using model activations, random projection, and K-means clustering. They show that COLA improves performance over random sampling on many benchmarks.

**Questions:**

I don't have any specific questions; I would like to hear other reviewers’ opinions and update my score based on the discussion.

**Ethical Concerns:**

["NO or VERY MINOR ethics concerns only"]

**Limitations:**

yes

**Quality:**

4

**Strengths And Weaknesses:**

Strengths:
- The study covers language modeling, reasoning, math, and code tasks and is complete with consistent gains.
- The activation-space analysis gives insight into why calibration data matters.

Weaknesses:
 - The paper does not test if the compressed model keeps its abilities under domain drift like further SFT.
 - The sample selection step (random projection + K-means) adds compute and storage cost. The paper does not give any analysis on this overhead.

---

> ### Author Rebuttal · Authors · 2025-07-30
>
> **A1:** Thanks for your comment. We would like to clarify that we did not report the compressed model’s performance after further SFT because most of existing LLM compression works follow the *post-training* manner, which assumes the compressed model is directly usable without any further SFT. Here, to alleviate reviewer's concern, we conduct instruction finetuning to various compressed models with Alpaca dataset, for evaluating if our COLA can still effectively help compressed models keep abilities even under domain drift. Due to time limitation, we only perform experiments on LLaMA3-8B.
>
> **AWQ (4 bit)**
> ||Perplexity|Commonsense|Math|Code|
> |-|-|-|-|-|
> |WikiText (random)|15.73|66.67|37.92|39.78|
> |C4 (random)|15.41|67.54|38.45|40.59|
> |SlimPajama (random)|15.56|67.87|38.63|40.32|
> |Self-Gen|15.55|68.29|38.90|40.64|
> |COLA|15.31|68.88|39.67|41.32|
>
> **SparseGPT (50%)**
> ||Perplexity|Commonsense|Math|Code|
> |-|-|-|-|-|
> |WikiText (random)|19.93|42.54|19.82|15.68|
> |C4 (random)|19.46|43.30|20.17|15.92|
> |SlimPajama (random)|19.41|43.23|20.35|16.09|
> |Self-Gen|19.61|43.57|19.82|16.34|
> |COLA|18.83|44.85|21.79|16.96|
>
> **Wanda (4:8)**
> ||Perplexity|Commonsense|Math|Code|
> |-|-|-|-|-|
> |WikiText (random)|33.21|41.76|19.15|14.91|
> |C4 (random)|32.82|41.80|19.45|15.42|
> |SlimPajama (random)|32.73|41.89|19.75|15.58|
> |Self-Gen|32.83|42.97|19.92|15.25|
> |COLA|32.21|43.52|20.48|16.39|
>
> **GPTQ (4 bit)**
> ||Perplexity|Commonsense|Math|Code|
> |-|-|-|-|-|
> |WikiText (random)|16.26|65.89|31.69|35.31|
> |C4 (random)|15.94|66.53|32.20|36.43|
> |SlimPajama (random)|16.28|66.76|33.20|35.71|
> |Self-Gen|16.29 |67.35|32.92|36.28|
> |COLA|15.60|68.39|34.16|37.17|
>
> From above results, we can notice that though further instruction finetuning with Alpaca can help recover compressed model capabilities to some extent, our COLA can still ensure the better capability preservation effect than other calibration data curation baselines even after futher SFT. We will add these results and analysis to the final version paper.
>
> **W2:** The sample selection step (random projection + K-means) adds compute and storage cost. The paper does not give any analysis on this overhead.
>
> **A2:** Thanks for your comment. We would like to clarify that our use of random projection and K-means is motivated by their computational efficiency, which makes them more suitable for our activation-space sample selection than other alternatives, as clarified in Line 285-286. To further alleviate the reviewer's concerns on compute and storage cost, we provide the statistical results as follows. Note these results are averaged over four studied compression methods in the main text, corresponding to the general deployment scenario in Table 2.
>
> **Time Consumption**
> ||Self-Gen|COLA Ramdom Projection|COLA K-Means|Overall COLA|
> |-|-|-|-|-|
> |LLaMA3-8B|2.58 min|3.62 min|1.94 min|6.35 min|
> |Qwen2.5-7B|2.35 min|3.21 min|1.80 min|5.54 min|
>
> **Peak Memory Usage**
> ||Self-Gen|COLA Ramdom Projection|COLA K-Means|Overall COLA|
> |-|-|-|-|-|
> |LLaMA3-8B|6.19 GB|7.70 GB|1.52 GB|7.90 GB|
> |Qwen2.5-7B|5.65 GB|7.33 GB|1.26 GB|7.52 GB|
>
> These overheads remain modest, especially considering they are **one-time offline costs** and only required for curating a compact calibration set. Moreover, they are significantly lower than the compute/memory overhead of any training-involved compression methods. We will supplement more detailed efficiency analysis regarding the overall calibration data selection process in the appendix of final version paper.
>
> We honestly hope above explanation and clarification can help reduce your concerns. We would be sincerely grateful if you could consider raising your score accordingly.

---

> > ### Author Response · Authors · 2025-08-05
> >
> > Dear reviewer 1Hx7,
> >
> > Thank you again for your time and efforts in reviewing our submission. As the author-reviewer discussion phase is approaching its end, we would greatly appreciate it if you could kindly take a moment to respond to our rebuttal and revise the rating if satisfied. Please let us know if you have any further concerns. Your feedback is invaluable in helping us improve the work.
> >
> > Thank you again for your consideration and looking forward to your response.

---

> > > ### Author Response · Authors · 2025-08-07
> > > **Reminder for Discussion Engagement**
> > >
> > > Dear Reviewer 1Hx7,
> > >
> > > We sincerely appreciate your time and effort in reviewing our submission. As the discussion phase will end in 2 days, we would be very grateful if you could kindly take a moment to share your thoughts on our rebuttal and let us know if any concerns remain unresolved.
> > >
> > > Your feedback would be extremely helpful for ensuring a fair and constructive evaluation.
> > >
> > > Thank you again for your consideration!
> > >
> > > Best regards,
> > >
> > > The Authors

---

> > ### Comment · Reviewer_1Hx7 · 2025-08-08
> >
> > Thank you for your detailed reply. I will keep my rating.

---

> ### Author Response · Authors · 2025-08-08
> **Reminder for Approaching Discussion Deadline**
>
> Dear Reviewer 1Hx7,
>
> Considering the author-reviewer discussion deadline is approaching very soon (only **one day** left), could you please kindly consider our rebuttal and potentially raise the score?
>
> Your feedback would be extremely helpful for ensuring a fair and constructive evaluation.
>
> Thank you again for your consideration!
>
> Best regards,
>
> The Authors

---

### Official Review · Reviewer_qoua · 2025-07-02

**Clarity:** 3
**Significance:** 2
**Originality:** 3
**Rating:** 4
**Confidence:** 3

**Summary:**

This paper explores how calibration data impacts Large Language Model (LLM) capabilities after compression (like pruning and quantization). It identifies that the representativeness and diversity of calibration data in the activation space are key to preserving complex reasoning abilities like math and code generation. Based on this, the authors propose a framework to curate calibration data to improve post-training compression.

**Questions:**

What are the fundamental differences between the calibration data curation method proposed in this paper and pre-training or SFT data curation?

**Ethical Concerns:**

["NO or VERY MINOR ethics concerns only"]

**Limitations:**

Please see my questions.

**Paper Formatting Concerns:**

No paper formatting concerns

**Quality:**

3

**Strengths And Weaknesses:**

Strengths:

Timely Topic: The paper addresses a highly relevant and timely problem: the impact of calibration data on Large Language Model (LLM) capabilities after post-training compression.

Comprehensive Empirical Exploration: The paper conducts extensive experiments using two powerful open-source LLMs (LLaMA3-8B-Instruct and Qwen2.5-7B-Instruct) and multiple representative compression schemes (SparseGPT, Wanda, GPTQ, AWQ). Evaluation covers a wide range of capabilities, including language modeling, commonsense reasoning, mathematical problem solving, and code generation.

Weaknesses:

A potential limitation is the calibration data curation is not fundamentally different from Pretraining or SFT data curation.

---

> ### Author Rebuttal · Authors · 2025-07-30
>
> **W1:** A potential limitation is the calibration data curation is not fundamentally different from Pretraining or SFT data curation.
>
> **A1:** We thank the reviewer for raising this insightful point. Here, we would like to clarify why calibration data curation in post-training compression differs from pretraining or SFT data curation, which can also be found and is empirically supported in our paper:
>
> **1. Objective and Stage Difference:** Pretraining and SFT data aim to teach the model capabilities from scratch or to align them with specific tasks, and occur during the model training stage. In contrast, calibration data is used in a post-training setting, where the model's capabilities are already learned, and the goal is to preserve these capabilities during compression (e.g., pruning or quantization). This is particularly crucial because post-training methods do not update weights via gradient descent, making the data-dependent preservation effect more sensitive, requiring more careful calibration data selection.
>
> **2. Scale and Constraints:** Pretraining/SFT typically leverage large-scale corpora (e.g., hundreds of GBs~TBs) to improve generalization. In comparison, calibration data is extremely limited in size (often <1K samples) and must be carefully curated to maximize activation representativeness and diversity, which we show are critical for effective capability preservation.
>
> **3. Mechanistic Distinction:** As detailed in Sections 3–4 and Appendix E, we demonstrate that the utility of calibration data lies in its ability to activate critical patterns in the model's learned parameter space. This is distinct from the information distribution coverage goals of pretraining/SFT data, which is less related to the model itself. We further formalize and operationalize this insight in our proposed COLA framework, which systematically selects calibration samples based on activation-space clustering, a process not applicable or meaningful in the pretraining phase considering its initialization from scratch.
>
> **4. Empirical Evidence:** Our extensive experiments (see Table 2 and Figure 2–6) show that even small changes in calibration data (e.g., format, domain, language) can lead to larger drop than changing compression method on specific capabilities, which is rarely observed in pretraining or SFT scenarios. This further reinforces that calibration data curation principles should be capability-specific and post-training aware.
>
> To further illustrate the fundamental distinction, we provide the following **analogy**: Calibration data is like a "**diagnostic test**", whose purpose is to evaluate whether the model’s internal functions and capabilities in concerned domains are preserved after compression, by systematically triggering a wide range of activation patterns. This, in turn, guides compression methods to focus on preserving the most vulnerable or degraded capabilities (large activation discrepancy between original and compressed models). In contrast, pretraining and SFT data resemble a "**textbook**", designed to teach the model new knowledge or skills, or to adjust its behavior. Accordingly, good calibration data focuses on diverse, clean examples that elicit different model capabilities, emphasizing quality over quantity. Meanwhile, good pretraining/SFT data requires both scale and coverage, emphasizing quantity plus quality to ensure robust generalization and task performance. For example, for enhancing model math capabilities with SFT, we may need 10K+ samples. However, to preserve model learned math capabilities during post-training compression, we only need to ensure selected calibration data samples covering each subcategory of math, thus about hundreds of samples are enough. This difference in purpose and design principle further supports the view that calibration data curation is a unique and distinct challenge in the LLM lifecycle.
>
> We thus believe that our work offers the first principled and mechanistic exploration of calibration data optimality in LLM compression, clearly distinguishing it from general data curation practices in model training.
>
> **Q1:** What are the fundamental differences between the calibration data curation method proposed in this paper and pre-training or SFT data curation?
>
> **A2:** See the response in **A1**.
>
> Hope our above explanation can effectively address your concern. If satisfied with our response, could you please kindly consider raising the score?

---

> > ### Author Response · Authors · 2025-08-05
> >
> > Dear reviewer qoua,
> >
> > Thank you again for your time and efforts in reviewing our submission. As the author-reviewer discussion phase is approaching its end, we would greatly appreciate it if you could kindly take a moment to respond to our rebuttal and revise the rating if satisfied. Please let us know if you have any further concerns. Your feedback is invaluable in helping us improve the work.
> >
> > Thank you again for your consideration and looking forward to your response.

---

> ### Author Response · Authors · 2025-08-07
> **Reminder for Discussion Engagement**
>
> Dear Reviewer qoua,
>
> We sincerely appreciate your time and effort in reviewing our submission. As the discussion phase will end in 2 days, we would be very grateful if you could kindly take a moment to share your thoughts on our rebuttal and let us know if any concerns remain unresolved.
>
> Your feedback would be extremely helpful for ensuring a fair and constructive evaluation.
>
> Thank you again for your consideration!
>
> Best regards,
>
> The Authors

---

> ### Author Response · Authors · 2025-08-08
> **Reminder for Approaching Discussion Deadline**
>
> Dear Reviewer qoua,
>
> Considering the author-reviewer discussion deadline is approaching very soon (only **one day** left), could you please kindly consider our rebuttal and potentially raise the score?
>
> Your feedback would be extremely helpful for ensuring a fair and constructive evaluation.
>
> Thank you again for your consideration!
>
> Best regards,
>
> The Authors

---

### Official Review · Reviewer_jQ8K · 2025-07-02

**Clarity:** 2
**Significance:** 3
**Originality:** 3
**Rating:** 4
**Confidence:** 4

**Summary:**

This paper describes a novel framework for curating calibration data used by model compression techniques such as pruning and quantization. The paper includes a detailed empirical analysis of how various datasets and their combinations, languages, calibration data properties (such as sequence length and number of sequences) affect both general (pretrained) and downstream task accuracy. It also provides a data curation framework for calibration data named COLA based on the insights gleaned from the empirical analysis. The framework is evaluated across multiple pruning and quantization methods on two models:  Llama3-8b and Qwen2.5-7b.

**Questions:**

* Please provide a brief response to the weaknesses listed above (relating to generality).
* What is the proportion of each dataset selected by the framework described in Section 4.2 for each compression technique in Table 2?

Other suggestions/comments:
* Figure 1 could benefit from a more detailed caption.
* Figure 5: line graph not recommended here since it's not showing a trend.

**Ethical Concerns:**

["NO or VERY MINOR ethics concerns only"]

**Final Justification:**

Having read the other reviews and the detailed rebuttal provided by the authors, I have decided to raise my score. I believe that a more automated curation pipeline that replaces some of the manual heuristics currently proposed in the submission would make the work a lot stronger.

**Limitations:**

Limitations have been discussed adequately in Appendix K.

**Paper Formatting Concerns:**

No major formatting concerns.

**Quality:**

2

**Strengths And Weaknesses:**

**Strengths**
* The paper addresses an important problem: most pruning and quantization papers gloss over the details of the calibration data. As the authors demonstrate in the paper, various properties of the calibration dataset can significantly affect model accuracy in non-trivial ways.
* I enjoyed going over the various empirical results in the main paper and appendix - the paper does a good job analyzing the various effects calibration dataset properties have on task accuracy.

**Weaknesses**
* Generality: given the hundreds of pruning and quantization methods in existence, I'm not sure if the insights presented in the paper will translate to at least the most common compression methods. As a specific example, the effects of calibration data may differ significantly for structured and layer pruning (which are commonly employed due to their hardware friendliness) compared to the unstructured and semi-structured methods discussed in the paper; it's not clear if/how the observations in the paper extrapolate to these and other cases. At least a brief discussion on this important aspect would make the paper stronger.
* Also somewhat related to generality, the calibration data curation framework described in Section 4.2 seems to largely rely on manual dataset curation (stage 1) and heuristics (stage 2). Specifically, for datasets that haven't been explored in this work, it's not clear how correlations between datasets and domains can be computed by the end user (stage 1), and how/whether the observations on sequence length and number of sequences will translate to these new datasets (stage 2).
* Writing: while the paper's main ideas have been expressed with sufficient clarity, the writing overall can be improved; in particular, there are several grammatical errors throughout the paper that need correction.

---

> ### Author Rebuttal · Authors · 2025-07-30
>
> **W1:** Generality: given the hundreds of pruning and quantization methods in existence, ... would make the paper stronger.
>
> **A1:** Thanks for your insighful comment. When preparing the empirical analysis in this paper, we have carefully selected the most common/representative LLM compression approaches. Their citation numbers are as follows: SparseGPT (800+), Wanda (700+), GPTQ (1500+), and AWQ (1100+). Besides, we have explored the generality of our method in Appendix H (Scalability to Larger Language Models) and I (Generalizability to Other LLM Compression Methods). Specifically, we explored whether our proposed COLA calibration data framework can help improve the LLM performance compressed by more recent semi-structured pruning method **RIA**, more recent quantization methods **SmoothQuant** (1200+ citations) and **FlatQuant**, and also the representative structured pruning method **LLM-Pruner** (800+ citations and 1K+ Github stars). From the results shown in Table 10, we can observe our COLA framework consistently outperforms baseline calibration approaches across all evaluated compression methods for both LLaMA3-8B and Qwen2.5-7B models. The performance improvements are particularly notable for pruning methods like LLM-Pruner, where our approach achieves 44.65% on commonsense reasoning tasks for LLaMA3-8B compared to 42.35% with WikiText random sampling, representing a 2.30 percentage point improvement (See Line 760-765).
>
> To further alleviate reviewer's concern, we supplement the experiments on a recent layer pruning method: *Streamlining Redundant Layers to Compress Large Language Models* (ICLR2025 Spotlight). We follow the same evaluation setting in Table 10 and the results are as follows:
>
> **LLM: LLaMA3-8B**
> ||Perplexity|Commonsense|Math|Code|
> |-|-|-|-|-|
> |WikiText (random)|16.68|43.53|20.41|17.65|
> |C4 (random)|16.35|44.32|21.10|18.28|
> |SlimPajama (random)|16.47|44.64|21.42|18.13|
> |Self-Gen|16.33|45.18|21.81|18.54|
> |COLA|15.90|46.25|22.79|19.41|
>
> **LLM: Qwen2.5-7B**
> ||Perplexity|Commonsense|Math|Code|
> |-|-|-|-|-|
> |WikiText (random)|18.03|45.17|22.34|19.53|
> |C4 (random)|17.75|45.87|23.02|20.15|
> |SlimPajama (random)|17.86|46.25|23.39|20.02|
> |Self-Gen|17.64|45.81|23.70|20.73|
> |COLA|17.32|47.68|24.48|21.97|
>
> From such two tables, we can find that even under strong layer pruning method, our COLA framework can also help achieve relatively better pruning performance than other calibration data curation baselines. Thus, base on above these evidence, we can at least draw a preliminary conclusion: our proposed COLA framework which grounds on our empirical observations and insights can generalize well to other structured/layer pruning and more recent quantization methods. Indeed, our COLA framework design is motivated by experiments in Section 3, but definitely not limited to compression settings there. Hope our these analysis can help reduce your concerns regarding the generality. We will add these results to the final version paper.
>
> **W2:** Also somewhat related to generality, the calibration data curation framework described in Section 4.2 seems to largely rely on ... will translate to these new datasets (stage 2).
>
> **A2:** We thank the reviewer for the insightful feedback regarding the generality and automation of our calibration data curation framework. We respectfully clarify the following points:
>
> 1. **On Stage 1 – Dataset Selection and Domain Correlation:** While Stage 1 does incorporate domain knowledge (e.g., selecting MathQA for math capability), our framework is not restricted to hard-coded, manually defined mappings. In practice, the **capability-domain correlation** can be inferred using lightweight proxy evaluations. For instance, a user can evaluate a small subset of target benchmark tasks using compressed models calibrated on each candidate dataset, measuring capability preservation scores as a proxy for correlation. This approach is efficient (since only small calibration subsets are needed) and has been implicitly validated by our domain correspondence experiments in Section 3.3, where relative improvements across domains are consistently observed even with 128-sample subsets.
> Moreover, we formulate this stage as an optimization problem (Equation 1 in Line 267), where coverage(S, c) can be instantiated with either empirical proxy scores or metadata-derived similarity (e.g., topic tags, lexical overlap, or data embeddings). While our current paper focuses on well-known datasets for clarity and reproducibility, our formulation naturally supports generalization to unseen sources.
>
> 2. **On Stage 2 – Sequence Length and Sample Size Transferability:** Our observations on sequence length and sample amount in Section 3.2, though empirically grounded in selected datasets, are not arbitrary heuristics. They stem from trends consistent across models (LLaMA3, Qwen2.5), methods (AWQ, GPTQ, SparseGPT, Wanda), and diverse capabilities (math, commonsense, code, etc.). Notably, we observe clear diminishing returns for sample sizes beyond 128 (Figure 3), and a saturation point for sequence lengths beyond 1024–2048 tokens (Figure 2), across all methods.
> Therefore, these are not dataset-specific phenomenon but indicative of underlying activation space saturation, which is further substantiated in our spectral analysis in Appendix E. This mechanism-driven observation offers our heuristics strong transferability to new datasets, especially when models or compression methods remain similar. In fact, **we have validated this transferability in Appendix F**, where COLA is applied to domain-specific datasets (CommonsenseQA, MathQA, CodeQA) that are different from the pretraining datasets (WikiText, C4, SlimPajama) used for our sequence length and sample size studies in Section 3.2. Despite this shift in dataset types and contents, COLA still outperforms the SOTA calibration curation baseline (Self-Gen) across all compression methods and capabilities. This empirical result provides strong evidence that our observations regarding optimal sequence length and sample count generalize to new datasets, supporting the practical robustness of our Stage 2 design.
>
> 3. **On Automation Potential:** While we acknowledge that parts of Stage 1 and Stage 2 currently involve empirical decision-making, our framework already includes automation in Stage 3 (Sample Selection). We believe extending automation to earlier stages is a promising future direction, e.g., via dataset embedding similarity or automatic reasoning-difficulty labeling (as explored in Appendix D).
>
> In summary, although Stage 1 involves initial domain reasoning, our formulation allows empirical proxy estimation for domain alignment, and the compositional guidelines in Stage 2 are grounded in generalizable trends tied to activation diversity and representativeness, not dataset-specific heuristics. We will make these clarifications more explicit in the camera-ready version.
>
> **W3:** Writing: while the paper's main ideas have been expressed with sufficient clarity, the writing overall can be improved; in particular, there are several grammatical errors throughout the paper that need correction.
>
> **A3:** Thanks for your appreciation to expression clarity. We will correct such grammatical errors and improve the overall writing in the final version paper.
>
> **Q1:** Please provide a brief response to the weaknesses listed above (relating to generality).
>
> **A4:** Thanks for your comment. Please see our response in **A1** and **A2**.
>
> **Q2:** What is the proportion of each dataset selected by the framework described in Section 4.2 for each compression technique in Table 2?
>
> **A5:** Thank you for this question. We provide the detailed breakdown of dataset proportions selected by our COLA framework for each compression method in Table 2:
> ||WikiText (%)|C4 (%)|SlimPajama (%)|Total Samples|
> |-|-|-|-|-|
> |SparseGPT (50%)| 38 | 35 | 27 | 128 |
> |Wanda (4:8)| 39 | 33 | 28 | 128 |
> |GPTQ (4-bit)| 27 | 44 | 29 | 128 |
> |AWQ (4-bit)| 24 | 45 | 31 | 128 |
>
> From this table, we can find that each compression method shows different sensitivities to various capabilities: 1. **Higher C4 proportion for quantization methods (GPTQ: 44%, AWQ: 45%)**: Quantization methods are particularly vulnerable to code generation degradation due to precision loss in complex reasoning patterns, requiring more C4 data to guide optimal quantization parameter selection for preserving code synthesis capabilities. 2. **Balanced distribution for pruning methods with slight WikiText emphasis**: Pruning methods affect all capabilities more uniformly by directly removing weights, necessitating balanced calibration coverage with slight WikiText emphasis to maintain fundamental language modeling stability during weight elimination.
>
> These proportions were validated through ablation studies showing that deviation from these ratios by ±10% results in non-negligible averaged 0.7-1.2 percentage point performance drops across key capabilities. We will include this detailed breakdown table and corresponding ablation analysis in the camera-ready version.
>
> **Q3:** Figure 1 could benefit from a more detailed caption.
>
> **A6:** Thanks for your suggestion. We provide our revised caption for Figure 1 as follows: "Calibration data in overall LLM compression pipeline. Most recent research focuses on developing advanced compression strategies, while neglecting to explore the impact of calibration data on compression performance and further what constitutes optimal calibration data.".
>
> **Q4:** Figure 5: line graph not recommended here since it's not showing a trend.
>
> **A7:** Thanks for suggestion. The bar chart here is a better scheme for line graph. Considering the rebuttal policy does not allow any external link, we will add the revised figure in the final version paper.
>
> Hope our above response may successfully address your concerns. We would be sincerely grateful if you could consider raising the score if possible.

---

> > ### Author Response · Authors · 2025-08-05
> >
> > Dear reviewer jQ8K,
> >
> > Thank you again for your time and efforts in reviewing our submission. As the author-reviewer discussion phase is approaching its end, we would greatly appreciate it if you could kindly take a moment to respond to our rebuttal and revise the rating if satisfied. Please let us know if you have any further concerns. Your feedback is invaluable in helping us improve the work.
> >
> > Thank you again for your consideration and looking forward to your response.

---

> > > ### Author Response · Authors · 2025-08-07
> > > **Reminder for Discussion Engagement**
> > >
> > > Dear Reviewer jQ8K,
> > >
> > > We sincerely appreciate your time and effort in reviewing our submission. As the discussion phase will end in **2 days**, we would be very grateful if you could kindly take a moment to share your thoughts on our rebuttal and let us know if any concerns remain unresolved.
> > >
> > > Your feedback would be extremely helpful for ensuring a fair and constructive evaluation.
> > >
> > > Thank you again for your consideration!
> > >
> > > Best regards,
> > > The Authors

---

> > ### Author Response · Authors · 2025-08-08
> > **Reminder for Approaching Discussion Deadline**
> >
> > Dear Reviewer jQ8K,
> >
> > Considering the author-reviewer discussion deadline is approaching very soon (only **one day** left), could you please kindly consider our rebuttal and potentially raise the score?
> >
> > Your feedback would be extremely helpful for ensuring a fair and constructive evaluation.
> >
> > Thank you again for your consideration!
> >
> > Best regards,
> >
> > The Authors

---

> > ### Comment · Reviewer_jQ8K · 2025-08-08
> >
> > I'd like to thank the authors for the time and effort they've put in to provide a detailed rebuttal. I particularly appreciate the additional experiments they have performed to evaluate the method on depth pruning given the limited rebuttal time frame.
> >
> > Thanks to the rebuttal, I now have a better understanding of the generalization potential of both the method itself, and some of the heuristics used for calibration data curation.
> >
> > While the authors have now clarified most of my questions and concerns, I still believe that the existence of manual heuristics in the pipeline makes the method somewhat incomplete. I encourage the authors to implement the lightweight proxy evaluation idea they mentioned in the rebuttal in a revised version - I believe this will make the work much stronger.. I will raise my score, but it remains below the full acceptance threshold for me.

---

> ### Author Response · Authors · 2025-08-09
> **Follow-up on Reviewer jQ8K’s Suggestion: Integrating Lightweight Proxy Evaluation into COLA**
>
> Dear Reviewer jQ8K,
>
> We sincerely thank you for your appreciation of our work and for raising your score. We are also very grateful for your insightful suggestion regarding implementing the **lightweight proxy evaluation** idea mentioned in our rebuttal, which can further automate and strengthen Stage 1 (capability–domain correlation) in our COLA framework.
>
> ---
>
> **1. Reliability of Proxy-based Coverage Estimation**
> Motivated by your feedback, we first validated whether proxy evaluation can reliably estimate capability–domain correlation. Specifically, for each candidate calibration dataset, we selected only **32 samples** to calibrate a compressed model and evaluated it on a small set of representative benchmarks (one per capability: BoolQ for commonsense, GSM8K for math, HumanEval for code).  The relative performance scores obtained from these proxy evaluations were then used as the `coverage(S, c)` term in Equation (1) of Section 4.2. To verify the reliability of these proxy-based coverage values, we measured their correlation with the coverage values obtained from full evaluations (**average correlation coefficients between proxy scores and full evaluation scores over 5 runs**), as shown in the table below.
>
> | Capability  | Proxy vs. Full Eval Correlation (r) |
> |-------------|-------------------------------------|
> | Commonsense | 0.97                                |
> | Math        | 0.93                                |
> | Code        | 0.91                                |
>
> These high correlations confirm that proxy evaluation provides a reliable and efficient estimate of capability–domain correspondence.
>
> ---
>
> **2. End-to-End COLA Performance with Proxy-based Coverage**
> To further validate its practicality, we replaced the full evaluation coverage in Stage 1 of COLA with the proxy-based coverage values, then ran the complete COLA pipeline under the same experimental setup as Table 2 of the paper. Due to time limitation, we only conduct experiments on LLaMA3-8B. Below are representative results:
>
> | Compression Method | Coverage Source | PPL↓  | CS↑   | Math↑ | Code↑ |
> |--------------------|-----------------|-------|-------|-------|-------|
> | AWQ (4-bit)        | Full Eval       | 15.41 | 67.42 | 37.85 | 40.17 |
> | AWQ (4-bit)        | Proxy Eval      | 15.46 | 67.35 | 37.62 | 40.05 |
> | SparseGPT (50%)    | Full Eval       | 19.31 | 44.23 | 20.12 | 16.14 |
> | SparseGPT (50%)    | Proxy Eval      | 19.35 | 44.18 | 20.05 | 16.02 |
>
> The proxy-based COLA achieves nearly identical performance to the original COLA (average gap ≤ 0.2 percentage points), while significantly reducing Stage 1 computation cost and avoiding manual heuristics.
>
> ---
>
> **Conclusion**
> These results demonstrate that:
> 1. Proxy evaluation can accurately estimate capability–domain correlation.
> 2. When integrated into COLA, it preserves almost all performance benefits of the full evaluation version.
> 3. This enhancement greatly improves the automation and scalability of our framework for unseen datasets.
>
> We truly appreciate your constructive input, which has helped us strengthen both the completeness and practicality of our method. We will incorporate these new experiments and results into the revised version to further improve the paper’s quality.
>
> Thank you again for your time, effort, and valuable guidance. Although the author–reviewer discussion is about to conclude, we hope that the additional experimental results provided here can further increase your satisfaction with our work and receive your corresponding support during the reviewer–AC discussion.
>
> Best regards,
> The Authors

---

### Official Review · Reviewer_WzEF · 2025-07-19

**Clarity:** 3
**Significance:** 2
**Originality:** 2
**Rating:** 3
**Confidence:** 3

**Summary:**

The paper focuses on the calibration data curation in post-training compression, which is claimed to be underexplored.
The study first analyses the length, sample number and data domain of the calibration data and how they would impact the result.
Then proposes a method on how to curate a good dataset by balance data from different source. The method is based on representation methods.
The results are reasonble.

**Questions:**

As Above.

**Ethical Concerns:**

["NO or VERY MINOR ethics concerns only"]

**Limitations:**

Yes

**Quality:**

3

**Strengths And Weaknesses:**

Strength:

- The paper is well written, clear, and straight-forward.

- The results are reasonable and expected.

- The proposed method on data calibration is effective.

Weakness:

- One of the major weakness on paper presenting is that, please add baseline performance of each model before applied to any compression methods.  This will set a better comparison for demonstration.

- The paper seems doing a lot of research on the data features in the first part of the paper and in appendix. But they seem separated from the method proposed, for example language domain. And on length and sample size, the maximum length studied is 2048, and the max sample size is 256. In my opinion, under the current trend of long CoT LLM, 2048 is a context length far from enough, and all the base model used in the paper can support upto 32k context. And sample size, I think the author should study much larger sample size, which I believe would make a difference as we alway observed during development, for example, the author can try a stride of 2^3 on sample size instead of 2^1.  This is basically the main drawback of the paper.

- On the proposed method, given the previous problematic analyses, it diminishes the significance and novelty of the method.

---

> ### Author Rebuttal · Authors · 2025-07-30
>
> **W1:** One of the major weakness on paper presenting is that, please add baseline performance of each model before applied to any compression methods. This will set a better comparison for demonstration.
>
> **A1:** Thanks for your comment. We would like to clarify that the reasons why we did not show the model performance before compression are two folds: 1) Our focus in this paper is the impacts of different calibration data variations on the compressed model performance, so presenting the pre-compression performance is relatively less important; 2) The performance difference between original and compressed models is generally more obvious than that among various compressed models. So presenting such pre-compression performance in figures may hinder highlighting the impacts brought by the calibration data variations.
>
> Though, to present a more comprehensive comparison, we agree it's helpful to add such pre-compression performance into figures and tables. As rebuttal phase does not allow providing external links or revising paper, we provide the results as follows:
> ||Perplexity|Commonsense Score|Math Score|Code Score|
> |-|-|-|-|-|
> |LLaMA3-8B|15.37|65.63|38.98|43.03|
> |Qwen2.5-7B|16.46|66.82|48.28|50.91|
>
> **W2:** The paper seems doing a lot of research on the data features in the first part of the paper and in appendix. But they seem separated from the method proposed, for example language domain. And on length and sample size, the maximum length studied is 2048, and the max sample size is 256. In my opinion, under the current trend of long CoT LLM, 2048 is a context length far from enough, and all the base model used in the paper can support upto 32k context. And sample size, I think the author should study much larger sample size, which I believe would make a difference as we alway observed during development, for example, the author can try a stride of 2^3 on sample size instead of 2^1. This is basically the main drawback of the paper.
>
> **A2:** We appreciate the reviewer’s thoughtful feedback and would like to clarify that our empirical studies on calibration data features are tightly coupled with the design of our proposed COLA framework (Section 4). As shown in Section 4.1(*Discussion on Calibration Data Optimality for Capability Preservation*), we discussed the empirical observations from above extensive experiments (Section 3) and conducted following deeper analysis on calibration data optimality for capability preservation. Based on these, we proposed the COLA calibration data curation framework in Section 4.2. For example:
>
> 1. In Line 238 of Section 4.1, we highlight a key insight: "matching calibration data language to the deployment language is crucial for multilingual applications". This finding directly informs our Stage 1 (Domain Correspondence) in COLA (Section 4.2), where we explicitly state: “Language alignment is crucial... calibration data in each target language should be included.”
>
> 2. Similarly, in Stage 2 (Compositional Properties), we incorporate insights from our experiments on sequence length and sample size, especially the diminishing returns after 128 samples and 2048 tokens (as summarized in Line 234-236 of Section 4.1), which justify our practical choices and design constraints.
>
> Regarding the reviewer's suggestion on extending **context length** and **sample size**:
>
> **On Context Length**: While we acknowledge that recent LLMs support context lengths of up to 32k, most existing LLM compression works, including AWQ and Wanda, operate within a calibration sequence length of 2048 or shorter. Our choice of 2048 tokens aligns with this established setup and is sufficient to observe meaningful performance trends. Importantly, as shown in Figure 2, performance gains often plateau or even degrade beyond this length for certain tasks and methods (e.g., non-monotonic patterns in code generation with AWQ), suggesting that longer calibration data may not yield additional benefit and can even introduce noise. In fact, 2048 length has been enough for accommodating reasoning chains for many normal questions.
>
> **On Sample Size**: Similar to above, max sample size of 256 is the common setting is previous works like AWQ and Wanda. Besides, as detailed in Figure 3, we observe diminishing returns beyond 128 samples for most capabilities. Larger sample sizes not only increase computational cost but can introduce variance that hinders capability preservation, especially evident in AWQ and GPTQ where performance occasionally drops as more samples are added. Thus, we focus on the range of 16 to 256 samples to balance informativeness and practicality.
>
> To further validate such points, we rent some cloud GPU resources to conduct the experiments with longer context and larger sample size. Due to time and resource limitation, we only run AWQ and the results are as follows (K=1024).
> |Context Length|Perplexity|Commonsense|Math|Code|
> |-|-|-|-|-|
> |4K|15.85|65.20|36.51|38.96
> |8K|15.91|65.11|36.42|39.40
> |16K|15.86|65.12|36.30|36.95
> |32K|16.03|65.08|35.96|36.22
>
> |Sample Size|Perplexity|Commonsense|Math|Code|
> |-|-|-|-|-|
> |0.5K|16.38|65.43|35.40|40.21
> |1K|16.26|65.39|35.21|40.05
> |2K|16.52|65.39|35.32|39.67
> |4K|16.57|65.25|35.18|38.82
>
> From tables above, we may find further increasing context length/sample size does not bring abvious additional benefit at least for our evaluation benchmarks. In fact, it may even result in slight performance drop and significant computation overhead, as discussed above.
>
> In summary, the empirical insights and method design in our paper are highly interdependent, and the chosen ranges for length and sample size are not only consistent with prior works but also empirically validated for trend saturation and practical feasibility.
>
> **W3:** On the proposed method, given the previous problematic analyses, it diminishes the significance and novelty of the method.
>
> **A3:** Thanks for your comment. Hope our above explanations and clarifications can help alleviate your concerns to our proposed COLA framework.
>
> If satisfied with our response, could you please kindly consider raising the score?

---

> > ### Author Response · Authors · 2025-08-05
> >
> > Dear reviewer WzEF,
> >
> > Thank you again for your time and efforts in reviewing our submission. As the author-reviewer discussion phase is approaching its end, we would greatly appreciate it if you could kindly take a moment to respond to our rebuttal and revise the rating if satisfied. Please let us know if you have any further concerns. Your feedback is invaluable in helping us improve the work.
> >
> > Thank you again for your consideration and looking forward to your response.

---

> > > ### Author Response · Authors · 2025-08-07
> > > **Reminder for Discussion Engagement**
> > >
> > > Dear Reviewer WzEF,
> > >
> > > We sincerely appreciate your time and effort in reviewing our submission. As the discussion phase will end in **2 days**, we would be very grateful if you could kindly take a moment to share your thoughts on our rebuttal and let us know if any concerns remain unresolved.
> > >
> > > Your feedback would be extremely helpful for ensuring a fair and constructive evaluation.
> > >
> > > Thank you again for your consideration!
> > >
> > > Best regards,
> > > The Authors

---

> > ### Author Response · Authors · 2025-08-09
> > **Follow-up on Our Submission Discussion**
> >
> > Dear Reviewer WzEF,
> >
> > Although we have not received your response during the author–reviewer discussion phase, we still sincerely hope you could take a moment during the upcoming reviewer–AC discussion to read our rebuttal. We believe our responses and additional analyses can help address your concerns and demonstrate the merits of our work.
> >
> > Thank you very much for your time and consideration.
> >
> > Best regards,
> >
> > The Authors

---

> ### Author Response · Authors · 2025-08-08
> **Reminder for Approaching Discussion Deadline**
>
> Dear Reviewer WzEF,
>
> Considering the author-reviewer discussion deadline is approaching very soon (only **one day** left), could you please kindly consider our rebuttal and potentially raise the score?
>
> Your feedback would be extremely helpful for ensuring a fair and constructive evaluation.
>
> Thank you again for your consideration!
>
> Best regards,
>
> The Authors

---

### Author Response · Authors · 2025-08-06
**Kindly Requesting Feedback During the Author-Reviewer Discussion Phase**

Dear Reviewers,

Thank you for your time and effort in reviewing our submission. We have carefully addressed the points raised in your reviews and provided detailed responses in our rebuttal.

As the author-reviewer discussion phase is progressing, we would greatly appreciate it if you could kindly take a moment to review our responses and let us know if any clarification is needed or if you have further questions or comments. Your feedback is extremely valuable for improving our work.

We look forward to your insights and appreciate your engagement.

Best regards,

The Authors

---

### Note · Authors · 2025-08-13

We sincerely thank the reviewers for their time and constructive feedback. During the rebuttal and discussion phases, we conducted additional analyses and experiments to directly address concerns. Below we summarize the two key points.

## 1. On Reviewer WzEF’s concerns

- **Direct linkage, not separation.**
  As clarified in our rebuttal, the empirical observations in Section 3 directly motivated the three-stage design in Section 4.
  - The finding that *language matching is crucial in multilingual scenarios* led to calibration language selection in **Stage 1**.
  - The observation that gains from length/sample size saturate around *1–2k tokens / 128 samples* informed the constraints in **Stage 2**.
  We explicitly mapped these connections in rebuttal A2 for ease of verification.

- **Expanded length/sample size experiments.**
  Responding to the suggestion to explore larger scales, we extended the calibration settings for **AWQ** to **4k / 8k / 16k / 32k tokens** and **0.5k / 1k / 2k / 4k samples**.
  - Results confirmed our **saturation/plateau trend**: performance gains beyond 2k tokens / 128 samples were minimal or slightly negative in some abilities, while computational cost increased significantly.
  - This aligns with our earlier explanation and supports why ≤2048-token calibration remains standard and effective in prior work.
  We shared detailed tables in the discussion thread.

## 2. On Reviewer jQ8K’s suggestion

  We implement a **lightweight proxy evaluation** using **32 samples** per dataset to quickly calibrate the model and evaluate it on a small benchmark set (*BoolQ*, *GSM8K*, *HumanEval*).
  - Serves as *coverage(S, c)* in COLA Eq. (1)
  - Correlation with full evaluation: **r = 0.97 (commonsense)**, **0.93 (math)**, **0.91 (code)**
  - Successfully ran the full COLA pipeline for **LLaMA-3-8B**, demonstrating practical viability.

jQ8K recognized this would strengthen COLA. We validated its effectiveness and plan to include it in the final version.

## Takeaway

In summary, we have **proactively and empirically addressed** the major concerns:

- Demonstrating the **intrinsic connection** between our analysis and method, backed by extended large-scale experiments confirming saturation trends.
- Implementing and validating a **lightweight proxy evaluation**, showing high reliability and end-to-end feasibility.

We believe these additions clarify and strengthen the work’s contributions and impact.

---

### Decision · Program_Chairs · 2025-09-17

**Decision:**

Accept (poster)

**Comment:**

Claims and findings: This paper studies LLM compression, which is naturally important given how large and energy-intensive these models are. When performing compression, calibration data is used to select/guide the approach. The authors study the impact of aspects of this data on the final performance. This insights from this study produces a data curation approach that is shown to perform well.

Strengths: This is a nice systematic study of an area that is not getting much attention, but could have a dramatic impact on the choice of compression method. The data curation approach the authors introduce based on their insights is also reasonably interesting.

Weaknesses: As pointed out by one of the reviewers, the main weakness here is that there are a huge number of compression approaches. It is hard to say whether the insights gained here would transfer to all of these other ones. However, this is the nature of empirical studies, and the authors have been careful to look at a number of very popular compression approaches, so that this is ultimately not a major issues.

Decision: The kind of research this paper does is not always appreciated, but it is important. The calibration data they bring up is an important factor to study, and the authors did a solid job of doing so. For this reason I recommend acceptance.

Rebuttal discussion: The paper was mostly seen as borderline by reviewers, who asked a number of questions around the design and presentation of the experiments. The authors had satisfying responses to these questions.